# [Solution] Matchmaker Paxos: A Reconfigurable Consensus Protocol

Michael Whittaker
*University of California, Berkeley*

Neil Giridharan
*University of California, Berkeley*

Adriana Szekeres
*University of Washington*

Joseph M. Hellerstein
*University of California, Berkeley*

Heidi Howard
*University of Cambridge*

Faisal Nawab
*University of California, Irvine*

Ion Stoica
*University of California, Berkeley*

## Abstract

State machine replication protocols, like MultiPaxos and Raft, are at the heart of numerous distributed systems. To tolerate machine failures, these protocols must replace failed machines with new machines, a process known as reconfiguration. Reconfiguration has become increasingly important over time as the need for frequent reconfiguration has grown. Despite this, reconfiguration has largely been neglected in the literature. In this paper, we present Matchmaker Paxos and Matchmaker MultiPaxos, a reconfigurable consensus and state machine replication protocol respectively. Our protocols can perform a reconfiguration with little to no impact on the latency or throughput of command processing; they can perform a reconfiguration in a few milliseconds; and they present a framework that can be generalized to other replication protocols in a way that previous reconfiguration techniques can not. We provide proofs of correctness for the protocols and optimizations, and present empirical results from an open source implementation showing that throughput and latency do not change significantly during a reconfiguration.

## 1 Introduction

Many distributed systems [4, 6, 7, 14, 16] rely on a state machine replication protocol, like MultiPaxos [17] or Raft [34], to keep multiple replicas of their data in sync. Over time, machines fail, and if too many machines in a state machine replication protocol fail, the protocol grinds to a halt. Thus, state machine replication protocols have to replace failed machines with new machines as the protocol runs, a process known as reconfiguration.

Reconfiguration is an essential component of state machine replication. It is not an optimization or an afterthought. Without a reconfiguration protocol in place, a state machine replication protocol will inevitably stop working; it's just a matter of when. Despite this, reconfiguration has largely been neglected by current academic literature. Researchers have invented dozens of state machine replication protocols, yet many papers either discuss reconfiguration briefly with no evaluation [31, 36–38], propose theoretically safe but inefficient reconfiguration protocols [18, 26], or do not discuss reconfiguration at all [2, 3, 19, 28, 29].

Ignoring reconfiguration has never been ideal, but we have largely been able to get away with it. Historically, state machine replication protocols were deployed on a fixed set of machines, and reconfiguration was used only to replace failed machines with new machines – an infrequent occurrence. This made it easy to leave reconfiguration out of sight, out of mind. Recently however, systems have become increasingly elastic, and the need for frequent reconfiguration has grown. These elastic systems don't just perform reconfigurations *reactively* when machines fail; they reconfigure *proactively*. For example, cloud databases can proactively request more resources to handle workload spikes, and orchestration tools like Kubernetes [15] are making it easier to build these types of elastic systems. Similarly, in environments with short-lived cloud instances—as with serverless computing and spot instances—and in mobile edge and Internet of Things settings, protocols must adapt to a changing set of machines much more frequently. This frequent need for reconfiguration makes it hard to ignore reconfiguration any longer.

In this paper, we present a reconfigurable consensus protocol and a reconfigurable state machine replication protocol: Matchmaker Paxos and Matchmaker MultiPaxos. In a nutshell, our protocols work by leveraging two key design ideas.

- The first is to *decouple reconfiguration* from the standard processing path. Many replication protocols [23, 26, 31, 34] have machines that are responsible for both processing commands and for orchestrating reconfigurations. By contrast, Matchmaker Paxos introduces a set of distinguished matchmaker machines that are solely responsible for managing reconfigurations and operate off of the critical path. These matchmakers act as a source of truth; they always know the current configuration.

- The second design point is to reconfigure across rounds, a technique known as *vertical reconfiguration* [22]. With vertical reconfiguration, every round of consensus can execute using a different configuration.

At the beginning of every round, the Paxos leader queries the matchmakers to discover the older configurations that were used in previous rounds, and it simultaneously sends the matchmakers the configuration it intends to use in the current round. In this way, the matchmakers act as a registry for configurations. Leaders simultaneously query the past and update the present. This matchmaking phase requires a single round trip of communication and happens rarely. We also introduce a number of novel protocol optimizations to perform the matchmaking completely off the critical path to avoid

degrading performance. Moreover, the protocol employs a garbage collection protocol to delete old configurations stored on the matchmakers. Our protocols have the following desirable properties.

**Little to No Performance Degradation.** Matchmaker MultiPaxos can perform a reconfiguration without significantly degrading the throughput or latency of processing client commands. For example, we show that reconfiguration has less than a 4% effect on the median of throughput and latency measurements (Section 7). Note that Matchmaker MultiPaxos is not the first protocol to achieve this [27].

**Quick Reconfiguration.** Matchmaker MultiPaxos can perform a reconfiguration quickly. Reconfiguring to a new set of machines takes one round trip of communication in the normal case (Section 4). Empirically, this requires only a few milliseconds within a single data center (Section 7). It takes slightly longer to shut down the old machines, but empirically this takes only five milliseconds within a data center (Section 7).

**Generality** Replication protocols based on classical Multi-Paxos assume a totally ordered log of chosen commands and reconfigure across log entries, known as *horizontal reconfiguration*. However, many state machine replication protocols do not replicate a log [2, 18, 31, 38, 39, 41]. These protocols cannot use horizontal reconfiguration. However, while none of these protocols have logs, they all have rounds and can implement vertical reconfiguration. This allows Matchmaker Paxos and Matchmaker MultiPaxos to serve as a foundation on top of which reconfiguration protocols can be built for these other non-log based protocols.

**Theoretical Insights.** Matchmaker Paxos generalizes Vertical Paxos [22], it is the first protocol to achieve the theoretical lower bound on Fast Paxos [19] quorum sizes, and it corrects errors in DPaxos [33] (Section 6).

**Proven Safe.** We describe Matchmaker Paxos and Matchmaker MultiPaxos precisely and prove that both are safe (Sections 3, 4, 5, A, B). Unfortunately, this is not often done for all reconfiguration protocols [30, 36–38].

## 2 Background

### 2.1 System Model

Throughout the paper, we assume an asynchronous network model in which messages can be arbitrarily dropped, delayed, and reordered. We assume machines can fail by crashing but do not act maliciously. We assume that machines operate at arbitrary speeds, and we do not assume clock synchronization. We assume a discovery service that nodes can use to find each other, but do not require that this service be strongly consistent. A node can safely communicate with outdated nodes. A system like DNS would suffice. Every protocol discussed in this paper assumes (for liveness) that at most $f$ machines will fail for some configurable $f$. All the protocols

discussed in this paper are safe, but due to the FLP impossibility result [10], none of the protocols are guaranteed to be fully live (unless the network is synchronous).

### 2.2 Paxos

A **consensus protocol** is a protocol that selects a single value from a set of proposed values. **Paxos** [17, 20] is one of the oldest and most popular consensus protocols. A Paxos deployment that tolerates $f$ faults consists of an arbitrary number of clients, at least $f + 1$ nodes called **proposers**, and $2f + 1$ nodes called **acceptors**, as illustrated in Figure 1. To reach consensus on a value, an execution of Paxos is divided into a number of rounds, each round having two phases: Phase 1 and Phase 2. Every round is orchestrated by a single predetermined proposer. The set of rounds can be any unbounded, totally ordered set. It is common to let the set of rounds be the set of lexicographically ordered integer pairs $(r, id)$ where $r$ is an integer and $id$ is a unique proposer id, where a proposer is responsible for executing every round that contains its id.

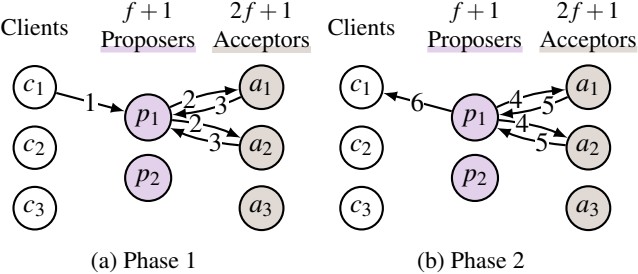

(a) Phase 1                           (b) Phase 2

Figure 1: Paxos communication diagram ($f = 1$).

When a proposer executes a round, say round $i$, it attempts to get some value $x$ chosen in that round. Paxos is a consensus protocol, so it must only choose a single value. Thus, Paxos must ensure that if a value $x$ is chosen in round $i$, then no other value besides $x$ can ever be chosen in any round less than $i$. This is the purpose of Paxos' two phases. In Phase 1 of round $i$, the proposer contacts the acceptors to (a) learn of any value that may have already been chosen in any round less than $i$ and (b) prevent any new values from being chosen in any round less than $i$. In Phase 2, the proposer proposes a value to the acceptors, and the acceptors vote on whether or not to choose it. In Phase 2, the proposer will only propose a value $x$ if it ensured through Phase 1 that no other value has been or will be chosen in a previous round.

More concretely, Paxos executes as follows, as illustrated in Figure 1. When a client wants to propose a value $x$, it sends $x$ to a proposer $p$. Upon receiving $x$, $p$ begins executing one round of Paxos, say round $i$. First, it executes Phase 1. It sends PHASE1A$\langle i \rangle$ messages to the acceptors. An acceptor ignores a PHASE1A$\langle i \rangle$ message if it has already received a message in a larger round. Otherwise, it replies with a PHASE1B$\langle i, vr, vv \rangle$ message containing the largest round $vr$ in which the acceptor

voted and the value it voted for, $vv$. If the acceptor hasn't voted yet, then $vr = -1$ and $vv = \mathsf{null}$. When the proposer receives PHASE1B messages from a majority of the acceptors, Phase 1 ends and Phase 2 begins.

At the start of Phase 2, the proposer uses the PHASE1B messages that it received in Phase 1 to select a value $x$ such that no value other than $x$ has been or will be chosen in any round less than $i$. Specifically $x$ is the vote value associated with the largest received vote round, or any value if no acceptor had voted (see [20] for details). Then, the proposer sends PHASE2A$\langle i, x \rangle$ messages to the acceptors. An acceptor ignores a PHASE2A$\langle i, x \rangle$ message if it has already received a message in a larger round. Otherwise, it votes for $x$ and sends back a PHASE2B$\langle i \rangle$ message to the proposer. If a majority of acceptors vote for the value, then the value is chosen, and the proposer informs the client. Proposer and acceptor pseudocode (with modifications for Matchmaker Paxos) are shown in Algorithm 3 and Algorithm 2.

## 2.3 Flexible Paxos

Paxos deploys a set of $2f + 1$ acceptors, and proposers communicate with at least a *majority* of the acceptors in Phase 1 and in Phase 2. **Flexible Paxos** [13] is a Paxos variant that eschews the notion of a *majority* for that of an arbitrary *quorum*. Specifically, Flexible Paxos introduces the notion of a **configuration** $C = (A; P1; P2)$. $A$ is a set of acceptors. $P1$ and $P2$ are sets of **quorums**, where each quorum is a subset of $A$. A configuration satisfies the property that every quorum in $P1$ (known as a **Phase 1 quorum**) intersects every quorum in $P2$ (known as a **Phase 2 quorum**). For a configuration to tolerate $f$ failures, there must exist some Phase 1 quorum and some Phase 2 quorum of non-failed machines despite an arbitrary set of $f$ failures.

Flexible Paxos is identical to Paxos with the exception that proposers now communicate with an arbitrary Phase 1 quorum in Phase 1 and an arbitrary Phase 2 quorum in Phase 2. In the remainder of this paper, we assume that all protocols operate using quorums from an arbitrary configuration rather than majorities from a fixed set of $2f + 1$ acceptors.

## 3 Matchmaker Paxos

We now present Matchmaker Paxos. To ease understanding, we first describe a simplified version of Matchmaker Paxos that is easy to understand but is also naively inefficient. We then upgrade the protocol to the complete, efficient version by way of a number of optimizations.

## 3.1 Overview and Intuition

Matchmaker Paxos is largely identical to Paxos. Like Paxos, a Matchmaker Paxos deployment includes an arbitrary number of clients, a set of at least $f + 1$ proposers, and some set

of acceptors, as illustrated in Figure 2. Paxos assumes that a *single, fixed* configuration of acceptors is used for every round. The big difference between Paxos and Matchmaker Paxos is that Matchmaker Paxos allows every round to have a *different* configuration of acceptors. Round 0 may use some configuration $C_0$, while round 1 may use some completely different configuration $C_1$. This idea was first introduced by Vertical Paxos [22].

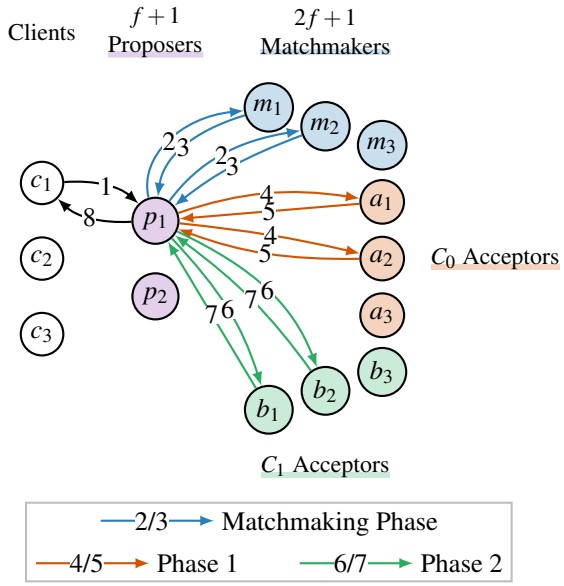

Figure 2: Matchmaker Paxos ($f = 1$).

Recall from Section 2 that a Paxos proposer in round $i$ executes Phase 1 in order to (1) learn of any value that may have been chosen in a round less than $i$ and (2) prevent any new values from being chosen in any round less than $i$. To do so, the proposer contacts the *fixed set* of acceptors. A Matchmaker Paxos proposer must also execute Phase 1 to establish that these two properties hold. The difference is that there is no longer a single fixed configuration of acceptors to contact. Instead, a Matchmaker Paxos proposer has to contact all of the configurations used in rounds less than $i$.

However, every round can use a different configuration of acceptors, so how does the proposer of round $i$ know which acceptors to contact in Phase 1? To resolve this question, a Matchmaker Paxos deployment also includes a set of $2f + 1$ **matchmakers**. The protocol executes as follows, as illustrated in Figure 2.

(1) A client proposes a value $x$ by sending it to a proposer ($p_1$ in Figure 2).

(2,3) When a proposer receives a value $x$, it begins executing the protocol in some round $i$. It selects a configuration $C_i$ and sends $C_i$ to the matchmakers. The matchmakers reply with the configurations used in previous rounds. We call this the **Matchmaking phase**. In Figure 2, the

proposer executes in round 1 and selects configuration $C_1$ consisting of acceptors $b_1$, $b_2$, and $b_3$. The matchmakers reply with the configuration $C_0$ consisting of acceptors $a_1$, $a_2$, and $a_3$.

(4,5) The proposer then executes Phase 1 of Paxos with the prior configurations that it received during the Matchmaking Phase. In Figure 2, the proposer executes Phase 1 with configuration $C_0$.

(6,7) The proposer then executes Phase 2 with the configuration $C_i$ to get the value $x$ chosen. In Figure 2, the proposer executes Phase 2 with configuration $C_1$.

(8) Finally, the proposer informs the client that $x$ was chosen.

At first, the extra round trip of communication with the matchmakers and the large number of configurations in Phase 1 make Matchmaker Paxos look slow. This is for ease of explanation. Later, we will eliminate these costs (Section 3.4 – Section 3.6).

## 3.2 Details

Every matchmaker maintains a log $L$ of configurations indexed by round. That is, $L[i]$ stores the configuration of round $i$. When a proposer receives a request $x$ from a client and begins executing round $i$, it first selects a configuration $C_i$ to use in round $i$. It then sends a MATCHA$\langle i, C_i \rangle$ message to all of the matchmakers.

When a matchmaker receives a MATCHA$\langle i, C_i \rangle$ message, it checks to see if it had previously received a MATCHA$\langle j, C_j \rangle$ message for some round $j \geq i$. If so, the matchmaker ignores the MATCHA$\langle i, C_i \rangle$ message. Otherwise, it inserts $C_i$ in log entry $i$ and computes the set $H_i$ of previous configurations in the log: $H_i = \{(j, C_j) \mid j < i, C_j \in L\}$. It then replies to the proposer with a MATCHB$\langle i, H_i \rangle$ message. Matchmaker pseudocode is given in Algorithm 1. An example execution of a matchmaker is illustrated in Figure 3.

---

**Algorithm 1** Matchmaker Pseudocode

**State:** a log $L$ indexed by round, initially empty
1: **upon** receiving MATCHA$\langle i, C_i \rangle$ from proposer $p$ **do**
2:    **if** $\exists$ a configuration $C_j$ in round $j \geq i$ in $L$ **then**
3:       ignore the MATCHA$\langle i, C_i \rangle$ message
4:    **else**
5:       $H_i \leftarrow \{(j, C_j) \mid C_j \in L\}$
6:       $L[i] \leftarrow C_i$
7:       send MATCHB$\langle i, H_i \rangle$ to $p$

---

When the proposer in round $i$ receives MATCHB$\langle i, H_i^1 \rangle$, ..., MATCHB$\langle i, H_i^{f+1} \rangle$ from $f + 1$ matchmakers, it computes $H_i = \cup_{j=1}^{f+1} H_i^j$. For example, with $f = 1$ and $i = 2$, if the proposer in round 2 receives MATCHB$\langle 2, \{(0, C_0)\} \rangle$ and

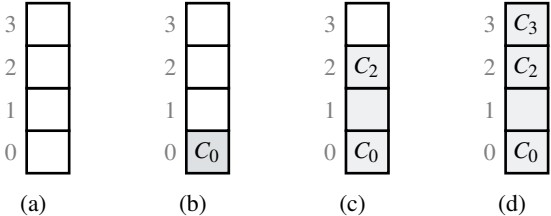

Figure 3: A matchmaker's log over time. (a) Initially, the matchmaker's log is empty. (b) Then, the matchmaker receives MATCHA$\langle 0, C_0 \rangle$. It inserts $C_0$ in log entry 0 and returns MATCHB$\langle 0, \emptyset \rangle$ since the log does not contain any configuration in any round less than 0. (c) The matchmaker then receives MATCHA$\langle 2, C_2 \rangle$. It inserts $C_2$ in log entry 2 and returns MATCHB$\langle 2, \{(0, C_0)\} \rangle$. (d) It then receives MATCHA$\langle 3, C_3 \rangle$, inserts $C_3$ in log entry 3, and returns MATCHB$\langle 3, \{(0, C_0), (2, C_2)\} \rangle$. At this point, if the matchmaker were to receive MATCHA$\langle 1, C_1 \rangle$, it would ignore it.

---

**Algorithm 2** Acceptor Pseudocode

**State:** the largest seen round $r$, initially $-1$
**State:** the largest round $vr$ voted in, initially $-1$
**State:** the value $vv$ voted for in round $vr$, initially null
1: **upon** receiving PHASE1A$\langle i \rangle$ from $p$ with $i > r$ **do**
2:    $r \leftarrow i$
3:    send PHASE1B$\langle i, vr, vv \rangle$ to $p$
4: **upon** receiving PHASE2A$\langle i, x \rangle$ from $p$ with $i \geq r$ **do**
5:    $r, vr, vv \leftarrow i, i, x$
6:    send PHASE2B$\langle i \rangle$ to $p$

---

MATCHB$\langle 2, \{(1, C_1)\} \rangle$, it computes $H_2 = \{(0, C_0), (1, C_1)\}$. Note that every round is statically assigned to a single proposer and that a proposer selects a single configuration for a round, so if two matchmakers return configurations for the same round, they are guaranteed to be the same.

The proposer then ends the Matchmaking phase and begins Phase 1. It sends PHASE1A messages to every acceptor in every configuration in $H_i$ and waits to receive PHASE1B messages from a Phase 1 quorum from every configuration. Using the previous example, the proposer sends PHASE1A messages to every acceptor in $C_0$ and $C_1$ and waits for PHASE1B messages from a Phase 1 quorum of $C_0$ and a Phase 1 quorum of $C_1$. The proposer then runs Phase 2 with $C_i$.

Acceptor and proposer pseudocode are shown in Algorithm 2 and Algorithm 3 respectively. To keep things simple, we assume that round numbers are integers, but generalizing to an arbitrary totally ordered set is straightforward. A Matchmaker Paxos acceptor is identical to a Paxos acceptor. A Matchmaker Paxos proposer is nearly identical to a Flexible Paxos proposer with the exception of the Matchmaking phase and the configurations used in Phase 1 and Phase 2. For clarity of exposition, we omit straightforward details surrounding re-sending dropped messages and nacking ignored messages.

**Algorithm 3** Proposer Pseudocode. Modifications to a Paxos proposer are underlined and shown in blue.

---

**State:** a value $x$, initially null
**State:** a round $i$, initially $-1$
**State:** the configuration $C_i$ for round $i$, initially null
**State:** the prior configurations $H_i$ for round $i$, initially null

1: **upon** receiving value $y$ from a client **do**
2:     $i \leftarrow$ next largest round owned by this proposer
3:     $x \leftarrow y$
4:     $C_i \leftarrow$ an arbitrary configuration
5:     send MATCHA$\langle i, C_i \rangle$ to all of the matchmakers

6: **upon** receiving MATCHB$\langle i, H_i^1 \rangle, \ldots,$ MATCHB$\langle i, H_i^{f+1} \rangle$ from $f+1$ matchmakers **do**
7:     $H_i \leftarrow \bigcup_{j=1}^{f+1} H_i^j$
8:     send PHASE1A$\langle i \rangle$ to every acceptor in $H_i$

9: **upon** receiving PHASE1B$\langle i, -, - \rangle$ from a Phase 1 quorum from every configuration in $H_i$ **do**
10:     $k \leftarrow$ the largest $vr$ in any PHASE1B$\langle i, vr, vv \rangle$
11:     **if** $k \neq -1$ **then**
12:         $x \leftarrow$ the corresponding $vv$ in round $k$
13:     send PHASE2A$\langle i, x \rangle$ to every acceptor in $C_i$

14: **upon** receiving PHASE2B$\langle i \rangle$ from a Phase 2 quorum **do**
15:     $x$ is chosen, inform the client

---

## 3.3 Proof of Safety

We now prove that Matchmaker Paxos is safe; i.e. every execution of Matchmaker Paxos chooses at most one value.

*Proof.* Our proof is based on the Paxos safety proof in [19]. We prove, for every round $i$, the statement $P(i)$: "if a proposer proposes a value $v$ in round $i$ (i.e. sends a PHASE2A message for value $v$ in round $i$), then no value other than $v$ has been or will be chosen in any round less than $i$." At most one value is ever proposed in a given round, so at most one value is ever chosen in a given round. Thus, $P(i)$ suffices to prove that Matchmaker Paxos is safe for the following reason. Assume for contradiction that Matchmaker Paxos chooses distinct values $x$ and $y$ in rounds $j$ and $i$ with $j < i$. Some proposer must have proposed $y$ in round $i$, so $P(i)$ ensures us that no value other than $y$ could have been chosen in round $j$. But, $x$ was chosen, a contradiction.

We prove $P(i)$ by strong induction on $i$. $P(0)$ is vacuous because there are no rounds less than 0. For the general case $P(i)$, we assume $P(0), \ldots, P(i-1)$. We perform a case analysis on the proposer's pseudocode (Algorithm 3). Either $k$ is $-1$ or it is not (line 11). First, assume it is not. In this case, the proposer proposes $x$, the value proposed in round $k$ (line 12). We perform a case analysis on round $j$ to show that no value other than $x$ has been or will be chosen in any round $j < i$. That is, we show $P(i)$.

**Case 1:** $j > k$**.** We show that no value has been or will be

chosen in round $j$. Recall that at the end of the Matchmaking phase, the proposer computed the set $H_i$ of prior configurations using responses from a set $M_i$ of $f+1$ matchmakers. Either $H_i$ contains a configuration $C_j$ in round $j$ or it doesn't.

First, suppose it does. Then, the proposer sent PHASE1A$\langle i \rangle$ messages to all of the acceptors in $C_j$. A Phase 1 quorum of these acceptors, say $Q$, all received PHASE1A$\langle i \rangle$ messages and replied with PHASE1B messages. Thus, every acceptor in $Q$ set its round $r$ to $i$, and in doing so, promised to never vote in any round less than $i$. Moreover, none of the acceptors in $Q$ had voted in any round greater than $k$. So, every acceptor in $Q$ has not voted and never will vote in round $j$. For a value $v'$ to be chosen in round $j$, it must receive votes from some Phase 2 quorum $Q'$ of round $j$ acceptors. But, $Q$ and $Q'$ necessarily intersect, so this is impossible. Thus, no value has been or will be chosen in round $j$.

Now suppose that $H_i$ does *not* contain a configuration for round $j$. $H_i$ is the union of $f+1$ MATCHB messages from the $f+1$ matchmakers in $M_i$. Thus, if $H_i$ does not contain a configuration for round $j$, then none of the MATCHB messages did either. This means that for every matchmaker $m \in M_i$, when $m$ received MATCHA$\langle i, C_i \rangle$, it did not contain a configuration for round $j$ in its log. Moreover, by processing the MATCHA$\langle i, C_i \rangle$ request, the matchmaker is guaranteed to never process a MATCHA$\langle j, C_j \rangle$ request in the future. Thus, every matchmaker in $M_i$ has not processed a MATCHA request in round $j$ and never will. For a value to be chosen in round $j$, the proposer executing round $j$ must first receive replies from $f+1$ matchmakers, say $M_j$, in round $j$. But, $M_i$ and $M_j$ necessarily intersect, so this is impossible. Thus, no value has been or will be chosen in round $j$.

**Case 2:** $j = k$**.** In a given round, at most one value is proposed, let alone chosen. $x$ is *the* value proposed in round $k$, so no other value could be chosen in round $k$.

**Case 3:** $j < k$**.** By induction, $P(k)$ states that no value other than $x$ has been or will be chosen in any round less than $k$. This includes round $j$.

Finally, if $k$ is $-1$, then we are in the same situation as in Case 1. No value has or will be chosen in a round $j < i$. $\square$

## 3.4 Garbage Collection (How)

We've discussed how a proposer can change its round and introduce a new configuration. Now, we explain how to shut down old configurations. At the beginning of round $i$, a proposer $p$ executes the Matchmaking phase and computes a set $H_i$ of configurations in rounds less than $i$. The proposer then executes Phase 1 with the acceptors in these configurations. Assume $H_i$ contains a configuration $C_j$ for a round $j < i$. If we prematurely shut down the acceptors in $C_j$, then proposer $p$ will get stuck in Phase 1, waiting for PHASE1B messages from a quorum of nodes that have been shut down. Therefore, we cannot shut down the acceptors in a configuration $C_j$ until we are sure that the matchmakers will never again return $C_j$

during the Matchmaking phase.

Thus, we extend Matchmaker Paxos to allow matchmakers to garbage collect configurations from their logs, ensuring that the garbage collected configurations will not be returned during any future Matchmaking phase. More concretely, a proposer $p$ can now send a GARBAGEA$\langle i \rangle$ command to the matchmakers informing them to garbage collect all configurations in rounds less than $i$. When a matchmaker receives a GARBAGEA$\langle i \rangle$ message, it deletes log entry $L[j]$ for every round $j < i$. It then updates a garbage collection watermark $w$ to the maximum of $w$ and $i$ and sends back a GARBAGEB$\langle i \rangle$ message to the proposer. See Algorithm 4.

---

**Algorithm 4** Matchmaker Pseudocode (with GC). Changes to Algorithm 1 are underlined and shown in blue.

**State:** a log $L$ indexed by round, initially empty
**State:** a garbage collection watermark $w$, initially 0
1: **upon** receiving GARBAGEA$\langle i \rangle$ from proposer $p$ **do**
2:     delete $L[j]$ for all $j < i$.
3:     $w \leftarrow \max(w, i)$
4:     send GARBAGEB$\langle i \rangle$ to $p$

5: **upon** receiving MATCHA$\langle i, C_i \rangle$ from proposer $p$ **do**
6:     **if** $i < w$ or $\exists\, C_j$ in round $j \geq i$ in $L$ **then**
7:         ignore the MATCHA$\langle i, C_i \rangle$ message
8:     **else**
9:         $H_i \leftarrow \{(j, C_j) \mid C_j \in L\}$
10:         $L[i] \leftarrow C_i$
11:         send MATCHB$\langle i, \underline{w}, H_i \rangle$ to $p$

---

We also update the Matchmaking phase in three ways. First, a matchmaker ignores a MATCHA$\langle i, C_i \rangle$ message if $i$ has been garbage collected (i.e. if $i < w$). Second, a matchmaker returns its garbage collection watermark $w$ in every MATCHB that it sends. Third, when a proposer receives MATCHB$\langle i, w_1, H_i^1 \rangle$, …, MATCHB$\langle i, w_{f+1}, H_i^{f+1} \rangle$ from $f + 1$ matchmakers, it again computes $H_i = \cup_{j=1}^{f+1} H_i^j$. It then computes $w = \max_{j=1}^{f+1} w_j$ and prunes every configuration in $H_i$ in a round less than $w$. In other words, if any of the $f + 1$ matchmakers have garbage collected round $j$, then the proposer also garbage collects round $j$.

Once a proposer receives GARBAGEB$\langle i \rangle$ messages from at least $f + 1$ matchmakers $M$, it is guaranteed that all future Matchmaking phases will not include any configuration in any round less than $i$. Why? Consider a future Matchmaking phase run with $f + 1$ matchmakers $M'$. $M$ and $M'$ intersect, so some matchmaker in the intersection has a garbage collection watermark at least as large as $i$. Thus, once a configuration has been garbage collected by $f + 1$ matchmakers, we can shut down the acceptors in the configuration.

## 3.5 Garbage Collection (When)

Once a configuration has been garbage collected, it is safe to shut it down, but when is it safe to garbage collect a configuration? It is not always safe. For example, if we prematurely garbage collect configuration $C_j$ in round $j$, a future proposer in round $i > j$ may not learn about a value $v$ chosen in round $j$ and then erroneously get a value other than $v$ chosen in round $i$. There are three situations in which it is safe for a proposer $p_i$ in round $i$ to issue a GARBAGEA$\langle i \rangle$ command. We explain the three situations and provide intuition on why they are safe. Later, we'll see that all three scenarios are important for Matchmaker MultiPaxos. See Section A for a safety proof.

**Scenario 1.** If the proposer $p_i$ gets a value $x$ chosen in round $i$, then it can safely issue a GARBAGEA$\langle i \rangle$ command. Why? When a proposer $p_j$ in round $j > i$ executes Phase 1, it will learn about the value $x$ and propose $x$ in Phase 2. But first, it must establish that no value other than $x$ has been or will be chosen in any round less than $j$. This is $P(j)$ from the safety proof in Section 3.3. The proposer $p_i$ already established this fact for all rounds less than $i$ (this is $P(i)$), so any communication with the configurations in these rounds is redundant. Thus, we can garbage collect them.

**Scenario 2.** If the proposer $p_i$ executes Phase 1 in round $i$ and finds $k = -1$ (see Algorithm 3), then it can safely issue a GARBAGEA$\langle i \rangle$ command. Recall that if $k = -1$, then no value has been or will be chosen in any round less than $i$. This situation is similar to Scenario 1. Any future proposer $p_j$ in round $j > i$ does not have to redundantly communicate with the configurations in rounds less than $i$ since $p_i$ already established that no value has been chosen in these rounds.

**Scenario 3.** If the proposer $p_i$ learns that a value $x$ has already been chosen and has been stored on $f + 1$ non-acceptor machines (e.g., $f + 1$ proposers), then the proposer can safely issue a GARBAGEA$\langle i \rangle$ command after it informs a Phase 2 quorum of acceptors in $C_i$ of this fact. Any future proposer $p_j$ in round $j > i$ will contact a Phase 1 quorum of $C_i$ and encounter at least one acceptor that knows the value $x$ has already been chosen. When this acceptor informs $p_j$ that a value $x$ has already been chosen, $p_j$ stops executing the protocol entirely and simply fetches the value $x$ from one of the $f + 1$ machines that store the value. Note that storing the value on $f + 1$ machines ensures that some machine will store the value despite $f$ failures. The decision of exactly which $f + 1$ machines is not important.

Later, we'll extend this garbage collection protocol to Matchmaker MultiPaxos (Section 4) and see empirically that matchmakers usually return just a single configuration (Section 7).

## 3.6 Optimizations

We now present a couple of protocol optimizations. First, note that a proposer can proactively run the Matchmaking

phase in round $i$ *before* it hears from a client. This is similar to proactively executing Phase 1, a standard optimization [12]. We call this optimization **proactive matchmaking**.

Second, assume that the proposer in round $i$ has executed the Matchmaking phase and Phase 1. Through Phase 1, it finds that $k = -1$ and thus learns that no value has been chosen in any round less than $i$ (see the safety proof above). Assume that before executing Phase 2 in round $i$, the proposer decides to perform a reconfiguration. To perform the reconfiguration, the proposer stops executing round $i$ and begins executing the next round $i+1$[1]. Typically to perform the reconfiguration, the proposer would have to execute the Matchmaking phase, Phase 1, and Phase 2 in round $i+1$. However, in this case, after executing the Matchmaking phase in round $i+1$, the proposer can skip Phase 1 and proceed directly to Phase 2. Why? The proposer established in round $i$ that no value has been or will be chosen in any round less than $i$. Moreover, because it did not run Phase 2 in round $i$, it also knows that no value has been or will be chosen in round $i$. Together, these imply that no value has been or will be chosen in any round less than $i+1$. Normally, the proposer would run Phase 1 in round $i+1$ to establish this fact, but since it has already established it, it can instead proceed directly to Phase 2. We call this optimization **Phase 1 bypassing**.

Phase 1 Bypassing depends on a proposer being the leader of round $i$ *and* the leader of the next round $i+1$. We can construct a set of rounds such that this is always the case. Let the set of rounds be the set of lexicographically ordered tuples $(r, id, s)$ where $r$ and $s$ are both integers and $id$ is a proposer id. A proposer is responsible for all the rounds that contain its id. With this set of rounds, the proposer $p$ in round $(r, p, s)$ always owns the next round $(r, p, s+1)$. For example given two proposers $a$ and $b$, we have the following ordering on rounds:

$$(0, a, 0) < (0, a, 1) < (0, a, 2) < (0, a, 3) < \cdots$$
$$(0, b, 0) < (0, b, 1) < (0, b, 2) < (0, b, 3) < \cdots$$
$$(1, a, 0) < (1, a, 1) < (1, a, 2) < (1, a, 3) < \cdots$$

We assume this round scheme throughout the rest of the paper. In the next section, we'll see that this optimization is essential for implementing Matchmaker MultiPaxos with good performance. Also note that this optimization is not particular to Matchmaker Paxos. Paxos and MultiPaxos can both take advantage of this optimization.

## 4  Matchmaker MultiPaxos

### 4.1  MultiPaxos

First, we summarize MultiPaxos. Whereas Paxos is a consensus protocol that agrees on a single value, **MultiPaxos** [17, 40]

---

is a **state machine replication protocol** that agrees on a sequence, or "log" of values. MultiPaxos manages multiple **replicas** of a state machine. Clients send state machine commands to MultiPaxos, MultiPaxos places the commands in a totally ordered log, and state machine replicas execute the commands in log order. By beginning in the same initial state and executing the same commands in the same order, all deterministic state machine replicas are kept in sync.

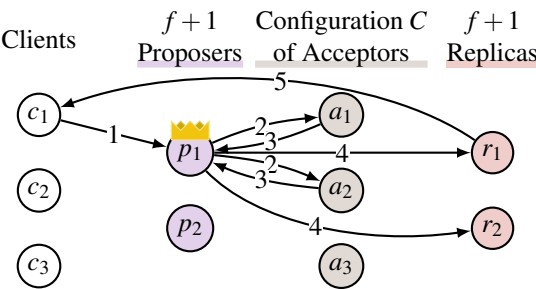

Figure 4: An example execution of MultiPaxos ($f = 1$). The leader is adorned with a crown.

To agree on a log of commands, MultiPaxos implements one instance of Paxos for every log entry. The $i$th instance of Paxos chooses the command in log entry $i$. More concretely, a MultiPaxos deployment that tolerates $f$ faults consists of an arbitrary number of clients, at least $f+1$ proposers, a configuration $C$ of acceptors which can tolerate $f$ failures, and at least $f+1$ replicas, as illustrated in Figure 4.

One of the proposers is elected leader in some round, say round $i$. We assume the leader knows that log entries up to and including log entry $k_c$ have already been chosen (e.g., by communicating with the replicas). We call this log entry the **commit index**. The leader then runs Phase 1 of Paxos in round $i$ for *every* log entry. Note that even though there are an infinite number of log entries larger than $k_c$, the leader can execute Phase 1 using a finite amount of information. In particular, the leader sends a single PHASE1A$\langle i \rangle$ message that acts as the PHASE1A message for every log entry larger than $k_c$. Also, an acceptor replies with a PHASE1B$\langle i, vr, vv \rangle$ message only for log entries in which the acceptor has voted. The infinitely many log entries in which the acceptor has not yet voted do not yield an explicit PHASE1B message.

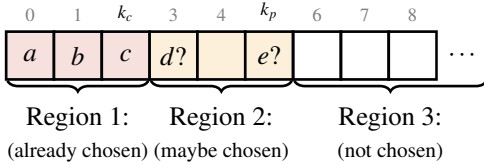

Figure 5: A leader's knowledge of the log after Phase 1.

The leader's knowledge about the log after Phase 1 can be characterized by the commit index $k_c$ and a **pending index**

---

[1]Note that given a round $i$, we denote the next largest round in the total ordered set of rounds $i+1$. We call this the "next round".

$k_p$ with $k_c \leq k_p$, as shown in Figure 5. The commit index and pending index divide the log into three regions: a prefix of chosen log entries (Region 1), a suffix of unchosen log entries (Region 3), and a middle region of pending log entries (Region 2). More specifically:

- **Region 1** $[0, k_c]$**:** The leader knows that a command has been chosen in every log entry less than or equal to $k_c$.

- **Region 3** $[k_p + 1, \infty)$**:** The leader knows that no command has been chosen (in any round less than $i$) in any log entry larger than $k_p$.

- **Region 2** $[k_c + 1, k_p]$**:** If there is a command that *may* have already been chosen, then it appears between $k_c$ and $k_p$. Region 2 may also contain some log entries in which the leader knows (from executing a previous round) that a value has already been chosen, and it may contain some log entries in which the leader knows (from counting votes in Phase 1) that no value has been chosen (we call these "holes").

After Phase 1, the leader sends a PHASE2A message for every unchosen log entry in Region 2, proposing a "no-op" command for the holes. Simultaneously, the leader begins accepting client requests. When a client wants to propose a state machine command, it sends the command to the leader. The leader assigns log entries to commands in increasing order, beginning at $k_p + 1$. It then runs Phase 2 of Paxos to get the command chosen in that entry in round $i$. Once the leader learns that a command has been chosen in a given log entry, it informs the replicas. Replicas insert chosen commands into their logs and execute the logs in prefix order, sending the results of execution back to the clients. This execution is illustrated in Figure 4.

It is critical to note that a leader performs Phase 1 of Paxos only once *per round*, not once *per command*. In other words, Phase 1 is not performed during normal operation. It is performed only when the leader fails and a new leader is elected in a larger round, an uncommon occurrence.

## 4.2 Matchmaker MultiPaxos

We first extend Matchmaker Paxos to Matchmaker Multi-Paxos with proactive matchmaking but without Phase 1 bypassing or garbage collection. We'll see how to incorporate these two momentarily in Section 4.4. The extension from Matchmaker Paxos to Matchmaker MultiPaxos is analogous to the extension of Paxos to MultiPaxos. Matchmaker MultiPaxos reaches consensus on a totally ordered log of state machine commands, one log entry at a time, using one instance of Matchmaker Paxos for every log entry.

More concretely, a Matchmaker MultiPaxos deployment consists of an arbitrary number of clients, at least $f + 1$ proposers, a set of $2f + 1$ matchmakers, a dynamic set of acceptors (one configuration per round which can tolerate $f$

failures), and a set of at least $f + 1$ state machine replicas. We assume, as is standard, that a leader election algorithm is used to select one of the proposers as a stable leader in some round, say round $i$. The leader selects a configuration $C_i$ of acceptors that it will use for *every* log entry. The mechanism by which the configuration is chosen is an orthogonal concern. A system administrator, for example, could send the configuration to the leader, or the configuration could be read from an external service. Throughout the paper, we do not depend any specific mechanism by which a configuration is chosen. We assume that proposers use some unspecified abstract process to select configurations.

The leader then executes the Matchmaking phase in the same way as in Matchmaker Paxos (i.e. it sends MATCHA$\langle i, C_i \rangle$ messages to the matchmakers and awaits MATCHB$\langle i, H_i \rangle$ responses). After the Matchmaking phase completes, the leader executes Phase 1 for *every* log entry. This is identical to MultiPaxos, except that the leader uses the configurations returned by the matchmakers rather than assuming a fixed configuration. Note that proactive matchmaking allows the leader to execute the Matchmaking phase and Phase 1 before receiving any client requests.

The leader then enters Phase 2 and operates exactly as it would in MultiPaxos. It executes Phase 2 with $C_i$ for the log entries in Region 2. Moreover, when it receives a state machine command from a client, it assigns the command a log entry in Region 3, runs Phase 2 with the acceptors in $C_i$, and informs the replicas when the command is chosen. Replicas execute commands in log order and send the results of executing commands back to the clients.

## 4.3 Discussion

To reconfigure from some old configuration $C_{old}$ in round $i$ to some new configuration $C_{new}$, the Matchmaker MultiPaxos leader of round $i$ simply advances to round $i + 1$ and selects the new configuration $C_{new}$. The new configuration is active immediately after the Matchmaking phase, a one round trip delay. Note that the acceptors in the new configuration $C_{new}$ do not have to undergo any sort of warm up or bootstrapping and do not have to contact any other acceptors in any other configuration.

The new configuration is active immediately, but it is not safe to deactivate the acceptors in the old configuration immediately, as we saw in Section 3.5. We extend Matchmaker Paxos's garbage collection to Matchmaker MultiPaxos momentarily.

Also note that Matchmaker MultiPaxos does *not* perform the Matchmaking phase or Phase 1 on the critical path of normal execution. Similar to how MultiPaxos executes Phase 1 only once per leader change (and not once per command), Matchmaker MultiPaxos runs the Matchmaking phase and Phase 1 only when a new leader is elected or when a leader changes its round (e.g., when a leader transitions from round

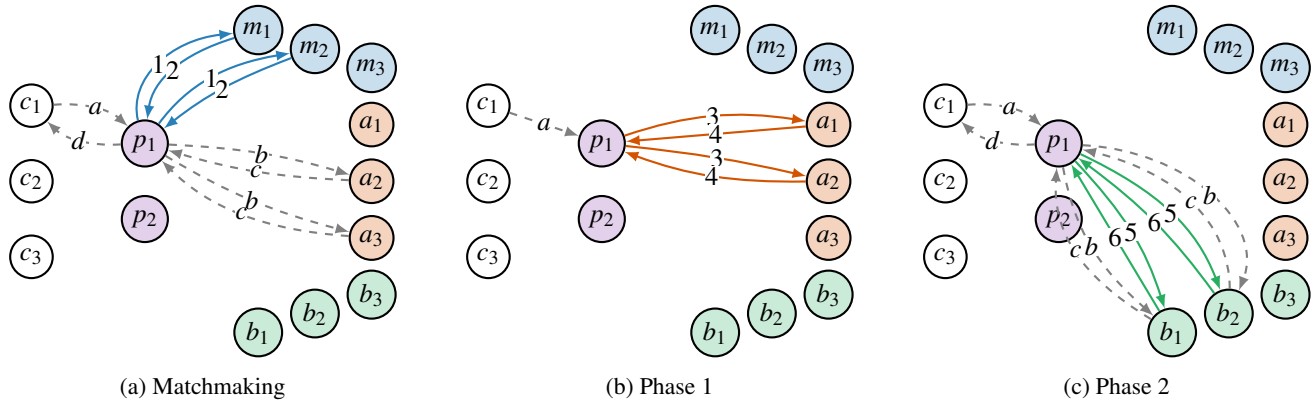

Figure 6: An example Matchmaker MultiPaxos reconfiguration without Phase 1 bypassing. The leader $p_1$ reconfigures from the acceptors $a_1$, $a_2$, $a_3$ to the acceptors $b_1$, $b_2$, $b_3$. Client commands are drawn as gray dashed lines. Note that every subfigure shows one phase of a reconfiguration using solid colored lines, but the dashed lines show the complete execution of a client request that runs concurrently with the reconfiguration. For simplicity, we assume that every proposer also serves as a replica.

$i$ to round $i+1$ as part of a reconfiguration). In the normal case (i.e. during Phase 2), Matchmaker MultiPaxos and MultiPaxos are identical, and Matchmaker MultiPaxos does not introduce *any* overheads. In the normal case, Matchmaker MultiPaxos deploys a single stable leader that changes rounds only to perform a reconfiguration. Changing from one leader to another only happens after a leader has failed.

Furthermore, configurations do not have to be unique across rounds. The leader in round $i$ is free to re-use a configuration $C_j$ that was used in some round $j < i$.

Finally, because Matchmaker MultiPaxos deploys more nodes than MultiPaxos, the mean time to failure is decreased, and it will take less time to reach $f$ failures. However, this mean time to failure is many orders of magnitude larger than the time required to perform a reconfiguration. As long as failed machines are replaced via reconfiguration in a reasonable amount of time, it is unlikely to experience $f$ or more failures.

### 4.4   Optimization

Ideally, Matchmaker MultiPaxos' performance would be unaffected by a reconfiguration. The latency of every client request and the protocol's overall throughput would remain constant throughout a reconfiguration. Matchmaker MultiPaxos as we've described it so far, however, does not meet this ideal. During a reconfiguration, a leader must temporarily stop processing client commands and wait for the reconfiguration to finish before resuming normal operation.

This is illustrated in Figure 6. Figure 6 shows a leader $p_1$ reconfiguring from a configuration of acceptors $C_{\text{old}}$ consisting of acceptors $a_1$, $a_2$, and $a_3$ in round $i$ to a new configuration of acceptors $C_{\text{new}}$ consisting of acceptors $b_1$, $b_2$, and $b_3$ in round $i+1$. While the leader performs the reconfiguration, clients continue to send state machine commands to the leader.

We consider such a command and perform a case analysis on when the command arrives at the leader to see whether or not the command has to be stalled.

**Case 1: Matchmaking (Figure 6a).** If the leader receives a command during the Matchmaking phase, then the leader can process the command as normal in round $i$ using the acceptors in $C_{\text{old}}$. Even though the leader is executing the Matchmaking phase in round $i+1$ and is communicating with the matchmakers, the acceptors in $C_{\text{old}}$ are oblivious to this and can process commands in Phase 2 in round $i$.

**Case 2: Phase 1 (Figure 6b).** If the leader receives a command during Phase 1, then the leader cannot process the command. It must delay the processing of the command until Phase 1 finishes. Here's why. Once an acceptor in $C_{\text{old}}$ receives a PHASE1A$\langle i+1\rangle$ message, it will reject any future commands in rounds less than $i+1$, so the leader is unable to send the command to $C_{\text{old}}$. The leader also cannot send the command to $C_{\text{new}}$ in round $i+1$ because it has not yet finished executing Phase 1.

**Case 3: Phase 2 (Figure 6c).** If the leader receives a command during Phase 2, then the leader can send the command to the new acceptors in $C_{\text{new}}$ in round $i+1$. This is the normal case of execution.

In summary, any commands received during Phase 1 of a reconfiguration are delayed. Fortunately, we can eliminate this problem by using Phase 1 bypassing. Consider a leader performing a reconfiguration from $C_i$ in round $i$ to $C_{i+1}$ in round $i+1$. At the end of the Matchmaking phase and at the beginning of Phase 1 (in round $i+1$), let $k$ be the largest log entry that the leader has assigned to a command. That is, all log entries after entry $k$ are empty. These log entries satisfy the preconditions of Phase 1 bypassing, so it is safe for the leader to bypass Phase 1 in round $i+1$ for these log entries in the following way. When a leader receives a command after the Matchmaking phase, it assigns the command a log entry

larger than $k$, skips Phase 1, and executes Phase 2 in round $i + 1$ with $C_{\text{new}}$ immediately.

With this optimization and the round scheme described in Section 3.6, no state machine commands are delayed. Commands received during the Matchmaking phase or earlier are chosen in round $i$ by $C_{\text{old}}$ in log entries up to and including $k$. Commands received during Phase 1, Phase 2, or later are chosen in round $i + 1$ by $C_{\text{new}}$ in log entries $k+1, k+2, k+3$, and so on. With this optimization Matchmaker MultiPaxos can be reconfigured with minimal performance degradation.

## 4.5   Garbage Collection

Recall that the Matchmaker MultiPaxos leader $p_i$ in round $i$ uses a single configuration $C_i$ for *every* log entry. The leader $p_i$ can safely issue a GARBAGEA$\langle i \rangle$ command to the matchmakers once it ensures that *every* log entry satisfies one of the three scenarios described in Section 3.5. Recall from Figure 5 that at the end of Phase 1 and at the beginning of Phase 2, the log can be divided into three regions. Each of the three garbage collection scenarios applies to one of the regions.

Scenario 2 applies to Region 3. These are the log entries for which $k = -1$. Scenario 1 applies to Region 2, once the leader has successfully chosen commands in all of the log entries in Region 2. Scenario 3 applies to Region 1 if we make the following adjustments. First, we deploy $2f + 1$ replicas instead of $f + 1$. Second, the leader ensures that the prefix of previously chosen log entries is stored on at least $f + 1$ of the $2f + 1$ replicas. Third, the leader informs a Phase 2 quorum of $C_i$ acceptors that these commands have been stored on the replicas. Every replica maintains a copy of the log of state machine commands and cannot discard a command after execution. The log must also be garbage collected over time, for example, by using snapshots [34]. Note that garbage collecting the log is an orthogonal (but also complicated) issue from garbage collecting old configurations. It must be done regardless of reconfigurations and is outside of the scope of this paper.

In summary, the leader $p_i$ of round $i$ executes as follows. It executes the Matchmaking phase to get the prior configurations $H_i$. It executes Phase 1 with the configurations in $H_i$. It enters Phase 2 and chooses commands in Region 2. It informs a Phase 2 quorum of $C_i$ acceptors once the commands in Region 1 have been stored on $f + 1$ replicas. It issues a GARBAGEA$\langle i \rangle$ command to the matchmakers and awaits $f + 1$ GARBAGEB$\langle i \rangle$ responses. At this point, all previous configurations can be shut down.

Note that the leader can begin processing state machine commands from clients as soon as it enters Phase 2. It does not have to stall commands during garbage collection. Note also that during normal operation, old configurations are garbage collected very quickly. In Section 7, we show that $H_i$ almost always contains a single configuration (i.e. $C_{i-1}$).

## 5   Reconfiguring Matchmakers

We've discussed how Matchmaker MultiPaxos allows us to reconfigure the set of acceptors. In this section, we discuss how to reconfigure proposers, replicas, and matchmakers (themselves).

Reconfiguring proposers and replicas is straightforward. In fact, Matchmaker MultiPaxos reconfigures proposers and replicas in exactly the same way as MultiPaxos [40], so we do not discuss it at length. In short, a proposer can be safely added or removed at any time. Replicas can also be safely added or removed at any time so long as we ensure that commands replicated on $f + 1$ replicas remain replicated on $f + 1$ replicas. This is a difficult, yet orthogonal challenge. Existing approaches can be adopted by Matchmaker MultiPaxos [35]. For performance, a newly introduced proposer should contact an existing proposer or replica to learn about the prefix of already chosen commands, and a newly introduced replica should copy the log from an existing replica.

Reconfiguring matchmakers is a bit more involved, but still relatively straightforward. Recall that proposers perform the Matchmaking phase only during a change in round. Thus, for the vast majority of the time—specifically, when there is a single, stable leader—the matchmakers are completely idle. This means that the way we reconfigure the matchmakers has to be safe, but it doesn't have to be efficient. The matchmakers can be reconfigured at any time between round changes without any impact on the performance.

Thus, we use the simplest approach to reconfiguration: we shut down the old matchmakers and replace them with new ones, making sure that the new matchmakers' initial state is the same as the old matchmakers' final state. More concretely, we reconfigure from a set $M_{\text{old}}$ of matchmakers to a new set $M_{\text{new}}$ as follows. First, a proposer (or any other node) sends a STOPA$\langle \rangle$ message to the matchmakers in $M_{\text{old}}$. When a matchmaker $m_i$ receives a STOPA$\langle \rangle$ message, it stops processing messages (except for other STOPA$\langle \rangle$ messages) and replies with STOPB$\langle L_i, w_i \rangle$ where $L_i$ is $m_i$'s log and $w_i$ is its garbage collection watermark. When the proposer receives STOPB messages from $f + 1$ matchmakers, it knows that the matchmakers have effectively been shut down. It computes $w$ as the maximum of every returned $w_i$. It computes $L$ as the union of the returned logs, and removes all entries of $L$ that appear in a round less than $w$. An example of this log merging is illustrated in Figure 7.

The proposer then sends $L$ and $w$ to all of the matchmakers in $M_{\text{new}}$. Each matchmaker adopts these values as its initial state. At this point, the matchmakers in $M_{\text{new}}$ *cannot* begin processing commands yet. Naively, it is possible that two different nodes could simultaneously attempt to reconfigure to two disjoint sets of matchmakers, say $M_{\text{new}}$ and $M'_{\text{new}}$.

To avoid this, every matchmaker in $M_{\text{old}}$ doubles as a Paxos acceptor. A proposer attempting to reconfigure to $M_{\text{new}}$ acts as a Paxos proposer and gets the value $M_{\text{new}}$ chosen by the

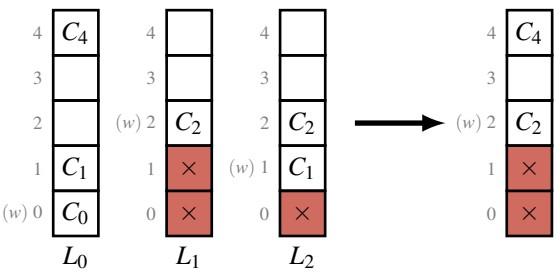

Figure 7: An example of merging three matchmaker logs ($L_0$, $L_1$, and $L_2$) during a matchmaker reconfiguration. Garbage collected log entries are shown in red.

matchmakers (which are acting as Paxos acceptors). Once $M_{\text{new}}$ is chosen, the proposer notifies the matchmakers in $M_{\text{new}}$ that the reconfiguration is complete and that they are free to start processing commands.

If a proposer contacts a stale set of matchmakers (e.g., $M_{\text{old}}$), the matchmakers inform the proposer of their successors (e.g., $M_{\text{new}}$). This newer set of matchmakers may also be stale, so the proposer repeatedly polls stale matchmakers until it finds the active set of matchmakers. In this way, the matchmakers form a chain, with each set of matchmakers pointing to its successor.

Before a set of matchmakers can be shut down, the identity of its successors must be persisted in some name service (e.g., DNS). Ideally for performance, the name service would always point to the active set of matchmakers, but this is not required for safety.

## 6   Theoretical Insights

**MultiPaxos**   To reconfigure from a set of nodes $N$ to a new set of nodes $N'$, a MultiPaxos leader gets the value $N'$ chosen in the log at some index $i$. All commands in the log starting at position $i + \alpha$ are chosen using the nodes in $N'$ instead of the nodes in $N$, where $\alpha$ is some configurable parameter. This protocol is called **Horizontal MultiPaxos**.

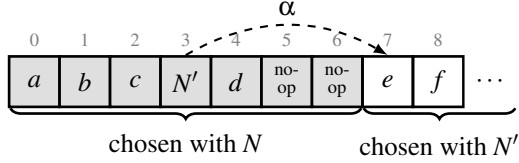

Figure 8: A MultiPaxos log during reconfiguration ($\alpha = 4$).

Matchmaker MultiPaxos has the following advantages over Horizontal MultiPaxos. First, the core idea behind Horizontal MultiPaxos seems simple, but the protocol has a number of hidden subtleties [27]. For example, a newly elected Horizontal MultiPaxos leader with a stale log may not know the latest

configuration of nodes. It may not even know which configuration of nodes to contact to learn the latest configuration of nodes. This makes it unclear when it is safe to shut down old configurations because a newly elected Horizontal Multi-Paxos leader can be arbitrarily out of date. These subtleties and the many others described in [27] makes Horizontal MultiPaxos significantly more complicated than it initially seems. Matchmaker Paxos addresses these subtleties directly. The matchmakers can always be used to learn the latest configuration, and our garbage collection protocol details exactly when and how to shut down old configurations safely.

Second, horizontal reconfiguration is not generally applicable. It is fundamentally incompatible with replication protocols that do not replicate a log. Moreover, researchers are finding that avoiding a log can often be advantageous [2, 8, 18, 31, 38, 39, 41]. For example, protocols like Generalized Paxos [18], EPaxos [31], Atlas [8], and Caesar [2] arrange commands in a partially ordered graph instead of a totally ordered log to exploit commutativity between commands. CASPaxos [38] maintains a single value, instead of a log or graph, for simplicity. Databases like TAPIR [41] avoid ordering transactions in a log for improved performance, and databases like Meerkat [39] do the same to improve scalability. Even some protocols with logs cannot use the ideas behind Horizontal MultiPaxos. For example, Raft cannot safely perform Horizontal MultiPaxos' reconfiguration [34].

Because these protocols do not replicate logs, they cannot use MultiPaxos' horizontal reconfiguration protocol. However, while none of the protocols replicate logs, *all* of them have rounds. This means that the protocols can either use Matchmaker Paxos directly, or at least borrow ideas from Matchmaker Paxos for reconfiguration. For example, we are developing a protocol called BPaxos that is an EPaxos [31] variant which partially orders commands into a graph. BPaxos is a modular protocol that uses Paxos as a black box subroutine. Due to this modularity, we can directly replace Paxos with Matchmaker Paxos to support reconfiguration. The same idea can also be applied to EPaxos. CASPaxos [38] is similar to Paxos and can be extended to Matchmaker CASPaxos in the same way we extended Paxos to Matchmaker Paxos. These are two simple examples, and we don't claim that extending Matchmaker Paxos to some of the other more complicated protocols is always easy. But, the universality of rounds makes Matchmaker Paxos an attractive foundation on top of which other non-log based protocols can build their own reconfiguration protocols.

One could argue that these other protocols are not used as much in industry, so it's not that important for them to have reconfiguration protocols, but we think the causation is in the reverse direction! Without reconfiguration, these protocols cannot be used in industry.

Third, optimizing Horizontal MultiPaxos is not easy. A MultiPaxos leader can process at most $\alpha$ unchosen commands at a time. This makes $\alpha$ an important parameter to tune. If we

set $\alpha$ too low, then we limit the protocol's pipeline parallelism and the throughput suffers. Note that a small $\alpha$ reduces the *normal case* throughput of Horizontal MultiPaxos, not just the throughput during reconfiguration. If we set $\alpha$ too high, then we have to wait a long time for a reconfiguration to complete. If we are reconfiguring because of a failed node, then we might have to endure a long reconfiguration with reduced throughput. Matchmaker MultiPaxos has no $\alpha$ parameter to tune. Note that Horizontal MultiPaxos can be implemented with an optimization in which we select a very large $\alpha$ and then get a sequence of $\alpha$ noops in the log to force a quick reconfiguration. This optimization helps avoid the difficulties of finding a good value of $\alpha$, but the optimization introduces a new set of subtleties into the protocol. For example, the leader cannot process client requests while it is executing Phase 2 for the $\alpha$ noops. The protocol has to implement additional mechanisms to avoid this one round trip stall.

Fourth, Horizontal MultiPaxos requires a Phase 1 and Phase 2 quorum of acceptors from an old configuration in order to perform a reconfiguration after a leader failure, but Matchmaker MultiPaxos only requires a Phase 1 quorum. Some read optimized MultiPaxos variants perform reads against Phase 1 quorums [5]. These protocols benefit from having very small Phase 1 quorums and very large Phase 2 quorums, requiring Horizontal MultiPaxos to contact far more nodes than Matchmaker MultiPaxos during a reconfiguration.

Finally, we clarify that if Horizontal MultiPaxos is implemented with all of its subtleties ironed out, is deployed with a good choice of $\alpha$, and is run with small Phase 2 quorums, then it can perform a reconfiguration without performance degradation. In this case, Horizontal MultiPaxos and Matchmaker MultiPaxos both reconfigure, in some sense, "optimally".

Horizontal MultiPaxos also has some advantages over Matchmaker MultiPaxos. For example, reconfiguring the set of matchmakers is simple, but it is still another reconfiguration protocol that has to be implemented which adds complexity to the system.

**Vertical Paxos**    Matchmaker MultiPaxos significantly improves the practicality of Vertical Paxos [22] in a number of ways. First, Vertical Paxos is a consensus protocol, not a state machine replication protocol, and it's not easy to extend Vertical Paxos' garbage collection protocol to a state machine replication protocol. Vertical Paxos garbage collects old configurations in situations similar to Scenario 1 and Scenario 2 from Section 3.5. It does not include Scenario 3. Without this, old configurations cannot be garbage collected, which means that it is never safe to shut down old configurations.

Second, Vertical Paxos requires an external master but does not describe how to implement the master in an efficient way. We could implement the master using another state machine replication protocol like MultiPaxos, but this would be both slow and overly complex. Plus, we would have to implement a reconfiguration protocol for the master as well.

Our matchmakers are analogous to the external master but show that such a master does not require a nested invocation of state machine replication.

Third, Vertical Paxos requires that a proposer execute Phase 1 in order to perform a reconfiguration. Thus, Vertical Paxos cannot be extended to MultiPaxos without causing performance degradation during reconfiguration. This is not the case for matchmakers thanks to Phase 1 bypassing.

Fourth, Vertical Paxos does not describe how proposers learn the configurations used in previous rounds and instead assumes that configurations are fixed in advance by an oracle. Matchmaker Paxos shows that this assumption is not necessary, as the matchmakers store every configuration.

**Fast Paxos**    Fast Paxos [19] is a Paxos variant that shaves off one network delay from Paxos in the best case, but can have higher delays if concurrently proposed commands conflict. While Paxos quorums consist of $f + 1$ out of $2f + 1$ acceptors, Fast Paxos requires larger quorums. Many protocols have reduced Fast Paxos quorum sizes a bit, but to date, Fast Paxos quorum sizes have remained larger than classic Paxos quorum sizes [8, 31]. Using matchmakers, we can implement Fast Paxos with a fixed set of $f + 1$ acceptors (and hence with $f + 1$-sized quorums). Specifically, we deploy Fast Paxos with $f + 1$ acceptors, with a single unanimous Phase 2 quorum, and with singleton Phase 1 quorums. A full description of the protocol and a proof of correctness is given in Section C.

**DPaxos**    DPaxos is a Paxos variant that allows every round to use a different subset of acceptors from some fixed set of acceptors. Matchmaker Paxos obviates the need for a fixed set of nodes. DPaxos' scope is limited to a single instance of consensus, whereas Matchmaker MultiPaxos shows how to efficiently reconfigure across multiple instances of consensus simultaneously. We also discovered that DPaxos' garbage collection algorithm is unsafe. Matchmaker MultiPaxos fixes the bug. See Section D for details.

# 7   Evaluation

We now evaluate Matchmaker MultiPaxos. Matchmaker MultiPaxos is implemented in Scala using the Netty networking library. We deployed Matchmaker MultiPaxos on m5.xlarge AWS EC2 instances within a single availability zone. We deploy Matchmaker MultiPaxos with $f = 1$, $f + 1$ proposers, $2f + 1$ acceptors, $2f + 1$ matchmakers, and $2f + 1$ replicas. For simplicity, every node is deployed on its own machine, but in practice, nodes can be physically co-located. In particular, any two logical roles can be placed on the same machine, so long as the two roles are not the same. For example, we can co-locate a leader, an acceptor, a replica, and a matchmaker, but we can't co-locate two acceptors (without reducing the fault tolerance of the system). For simplicity, we deploy

Table 1: Figure 9 median, interquartile range, and standard deviation of latency and throughput.

| | Latency (ms) | | | | | |
| | 1 Client | | 4 Clients | | 8 Clients | |
| | 0s-10s | 10s-20s | 0s-10s | 10s-20s | 0s-10s | 10s-20s |
|---|---|---|---|---|---|---|
| median | 0.292 | 0.287 | 0.317 | 0.321 | 0.398 | 0.410 |
| IQR | 0.040 | 0.026 | 0.029 | 0.036 | 0.036 | 0.039 |
| stdev | 0.114 | 0.085 | 0.076 | 0.081 | 0.089 | 0.305 |

| | Throughput (commands/second) | | | | | |
| | 1 Client | | 4 Clients | | 8 Clients | |
| | 0s-10s | 10s-20s | 0s-10s | 10s-20s | 0s-10s | 10s-20s |
|---|---|---|---|---|---|---|
| median | 2,995 | 3,177 | 11,874 | 11,478 | 19,146 | 18,446 |
| IQR | 152 | 53 | 175 | 145 | 140 | 373 |
| stdev | 157 | 111 | 298 | 307 | 358 | 520 |

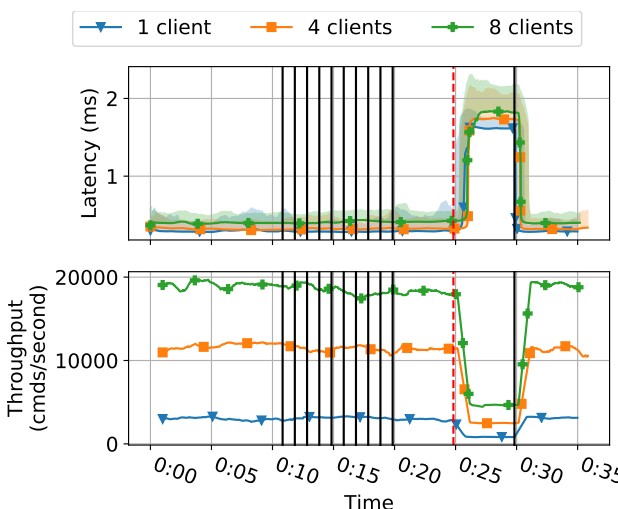

Figure 9: Matchmaker MultiPaxos' latency and throughput ($f = 1$). Median latency is shown using solid lines, while the 95% latency is shown as a shaded region above the median latency. The vertical black lines show reconfigurations. The vertical dashed red line shows an acceptor failure.

Matchmaker MultiPaxos with a trivial no-op state machine in which every state machine command is a one byte no-op. All of our results generalize to more complex state machines as well (the choice of state machine is orthogonal to reconfiguration).

## 7.1 Reconfiguration

**Experiment Description.** We run a benchmark with 1, 4, and 8 clients. Every client executes in a closed loop. It repeatedly proposes a state machine command, waits to receive a response, and then immediately proposes another command. This model is standard for state machine replication protocols [25, 31, 36] and aligns with the definitions surrounding linearizability [11]. Every benchmark runs for 35 seconds. During the first 10 seconds, we perform no reconfigurations. From 10 seconds to 20 seconds, the leader reconfigures the set of acceptors once every second. In practice, we would reconfigure much less often. This is a worst case stress test for Matchmaker MultiPaxos. For each of the ten reconfigurations, the leader selects a random set of $2f + 1$ acceptors from a pool of $2 \times (2f + 1)$ acceptors. At 25 seconds, we fail one of the acceptors. 5 seconds later, the leader performs a reconfiguration to replace the failed acceptor. The delay of 5 seconds is completely arbitrary. The leader can reconfigure sooner if desired.

We also perform this experiment with an implementation of MultiPaxos with horizontal reconfiguration. As with Matchmaker MultiPaxos, we deploy MultiPaxos with $f + 1$ proposers, $2f + 1$ acceptors, and $2f + 1$ replicas. We set $\alpha$ to 8. Because $\alpha$ is equal to the number of clients, MultiPaxos never stalls because of an insufficiently large $\alpha$. We do not implement the noop optimization.

**Results.** The latency and throughput of Matchmaker MultiPaxos are shown in Figure 9. Throughput and latency are both computed using sliding one second windows. Median latency

is shown using solid lines, while the 95% latency is shown as a shaded region above the median latency. The black vertical lines denote reconfigurations, and the red dashed vertical line denotes the acceptor failure.

The medians, interquartile ranges (IQR), and standard deviations of the latency and throughput (a) during the first 10 seconds and (b) between 10 and 20 seconds are shown in Table 1. Figure 12 includes violin plots of the same data. The white circles show the median values, while the thick black rectangles show the 25th and 75th percentiles. For latency, reconfiguration has little to no impact (roughly 2% changes) on the medians, IQRs, or standard deviations. The one exception is that the 8 client standard deviation is significantly larger. This is due to a small number of outliers. Reconfiguration has little impact on median throughput, with all differences being statistically insignificant. The IQRs and standard deviations sometimes increase and sometimes decrease. The IQR is always less than 1% of the median throughput, and the standard deviation is always less than 4%.

For every reconfiguration, the new acceptors become active within a millisecond. The old acceptors are garbage collected within five milliseconds. This means that only one configuration is ever returned by the matchmakers. We implement Matchmaker MultiPaxos with an optimization called thriftiness [31]—where PHASE2A messages are sent to a randomly selected Phase 2 quorum—so the throughput and latency expectedly degrade after we fail an acceptor. After we replace the failed acceptor, throughput and latency return to normal within two seconds.

The latency and throughput of MultiPaxos is shown in Fig-

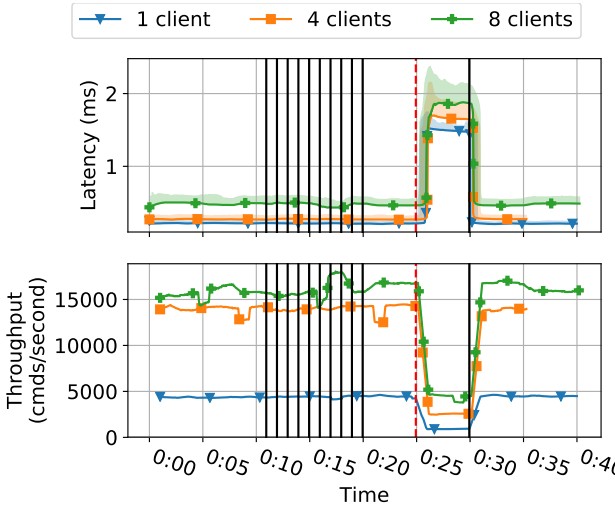

Figure 10: The latency and throughput of MultiPaxos with horizontal reconfiguration ($f = 1$).

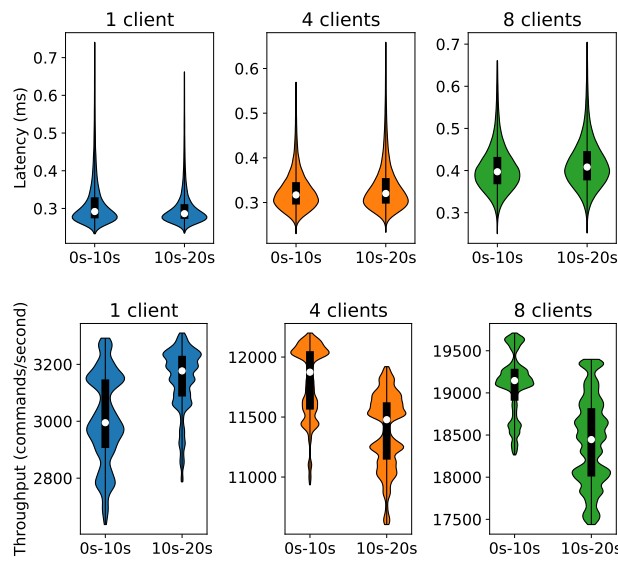

Figure 11: Violin plots of Figure 9 latency and throughput during the first 10 seconds and between 10 and 20 seconds.

ure 10. As with Matchmaker MultiPaxos, MultiPaxos can perform a horizontal reconfiguration without any performance degradation. The difference in absolute throughput between the two protocols is due to minor implementation differences, but the variance in throughput (rather than the throughput itself) is what is important for this evaluation. We include the comparison to MultiPaxos for the sake of having some baseline against which we can compare Matchmaker Multi-Paxos, but the comparison is shallow. For this reason, we do not elaborate on the results much.

While Matchmaker MultiPaxos does provide performance benefits over MultiPaxos' and Raft's reconfiguration protocols, our goal is not to replace these protocols. Rather, there are dozens of other state machine replication protocols (e.g., EPaxos [31], CASPaxos [38], Caesar [2], Atlas [8]) and distributed databases (e.g., TAPIR [41], Janus [32], Ocean Vista [9]) that do not have any reconfiguration protocol and cannot use the existing reconfiguration protocols from Mul-tiPaxos or Raft. Our hope is that the ideas in Matchmaker MultiPaxos can be used to implement reconfiguration protocols for these other systems. For this reason, it is difficult to compare Matchmaker MultiPaxos against some existing baseline because they simply do not exist.

**Summary.** This experiment confirms that Matchmaker MultiPaxos's throughput and latency remain steady even during abnormally frequent reconfiguration. Moreover, it confirms that Matchmaker MultiPaxos can reconfigure to a new set of acceptors and retire the old set of acceptors on the order of milliseconds.

## 7.2 Leader Failure

**Experiment Description.** We deploy Matchmaker Multi-Paxos exactly as before. Now, each benchmark runs for 20 seconds. During the first 7 seconds, there are no reconfigurations and no failures. At 7 seconds, we fail the leader. 5 seconds later, a new leader is elected and resumes normal operation. The 5 second delay is arbitrary; a new leader could be elected quicker if desired.

**Results.** The latency and throughput of the benchmarks are shown in Figure 13. During the first 7 seconds, throughput and latency are both stable. When the leader fails, the throughput expectedly drops to zero. The throughput and latency return to normal within two seconds after a new leader is elected. The results for the same experiment, repeated with Horizontal MultiPaxos, are shown in Figure 14.

**Summary.** This experiment confirms that the extra latency of the Matchmaker phase during a leader change is negligible.

## 7.3 Matchmaker Reconfiguration

**Experiment Description.** We deploy Matchmaker Multi-Paxos as above. We again run three benchmarks with 1, 4, and 8 clients. Each benchmark runs for 40 seconds. During the first 10 seconds, there are no reconfigurations and no failures. Between 10 and 20 seconds, the leader reconfigures the set of matchmakers once every second. Every reconfiguration randomly selects $2f + 1$ matchmakers from a set of $2 \times (2f + 1)$ matchmakers. At 25 seconds, we fail a matchmaker. At 30 we perform a matchmaker reconfiguration to replace the failed matchmaker. At 35 seconds, we reconfigure the acceptors.

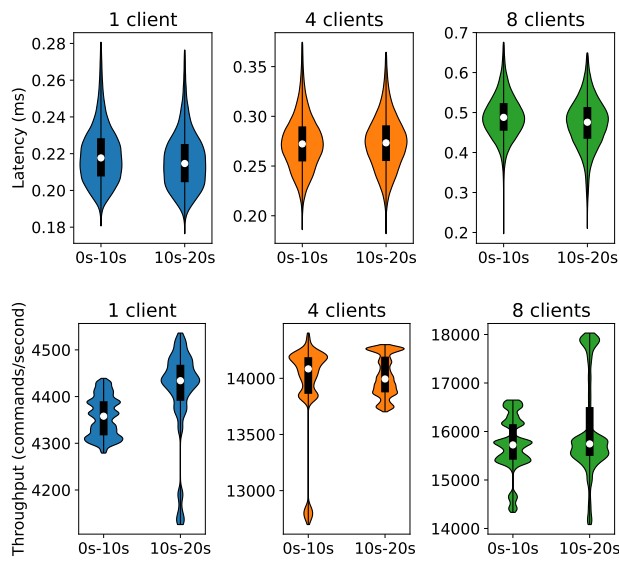

Figure 12: Violin plots of Figure 10 latency and throughput during the first 10 seconds and between 10 and 20 seconds.

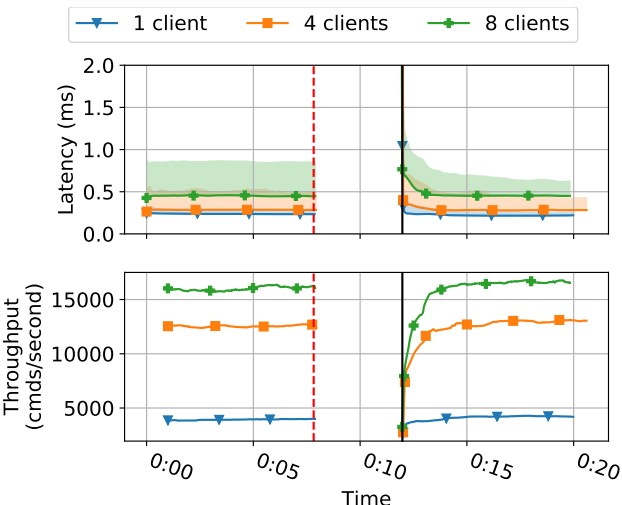

Figure 13: Matchmaker MultiPaxos' latency and throughput ($f = 1$). The dashed red line denotes a leader failure.

**Results.** The latency and throughput of Matchmaker MultiPaxos are shown in Figure 15. The latency and throughput of the protocol remain steady through the first ten matchmaker reconfigurations, through the matchmaker failure and recovery, and through the acceptor reconfiguration. This is confirmed by the medians, IQRs, and standard deviations of the latency and throughput during the first 10 seconds and between 10 and 20 seconds, which are shown in Table 2.

**Summary.** This benchmark confirms that matchmakers are off the critical path. The latency and throughput of Matchmaker MultiPaxos remains steady during a matchmaker reconfiguration and matchmaker failure. Moreover, a matchmaker reconfiguration does not affect the performance of subsequent acceptor reconfigurations.

Table 2: Figure 15 median, interquartile range, and standard deviation of latency and throughput.

| | Latency (ms) | | | | | |
|---|---|---|---|---|---|---|
| | 1 Client | | 4 Clients | | 8 Clients | |
| | 0s-10s | 10s-20s | 0s-10s | 10s-20s | 0s-10s | 10s-20s |
| median | 0.297 | 0.292 | 0.314 | 0.313 | 0.404 | 0.398 |
| IQR | 0.032 | 0.024 | 0.031 | 0.030 | 0.035 | 0.028 |
| stdev | 0.077 | 0.061 | 0.093 | 0.098 | 0.383 | 0.067 |

| | Throughput (commands/second) | | | | | |
|---|---|---|---|---|---|---|
| | 1 Client | | 4 Clients | | 8 Clients | |
| | 0s-10s | 10s-20s | 0s-10s | 10s-20s | 0s-10s | 10s-20s |
| median | 3019 | 3147 | 11631 | 11726 | 18569 | 19248 |
| IQR | 41 | 51 | 140 | 145 | 391 | 71 |
| stdev | 66 | 72 | 250 | 231 | 478 | 159 |

## 8 Related Work

**SMART.** SMART [27] is a reconfiguration protocol that resolves many ambiguities in MultiPaxos' horizontal approach (e.g., when it is safe to retire old configurations). Like MultiPaxos' horizontal reconfiguration protocol, SMART can reconfigure a protocol with minimal performance degradation. SMART differs from Matchmaker Paxos in a number of ways. First, like MultiPaxos' horizontal reconfiguration protocol, SMART is fundamentally log based and is therefore incompatible with many sophisticated state machine replication protocols. Second, SMART assumes that acceptors and replicas are always co-located. This prevents us from reconfiguring the acceptors without reconfiguring the replicas. This is not ideal since we can reconfigure an acceptor without copying any state, but must transfer logs from an old replica

to a new replica. SMART's garbage collection also has higher latency that Matchmaker Paxos' garbage collection. For Scenario 3, Matchmaker Paxos proposers wait until a prefix of the log is stored on $f + 1$ replicas. SMART waits for the prefix of the log to be executed and snapshotted by $f + 1$ replicas.

**Cheap Paxos.** Cheap Paxos [24] is a MultiPaxos variant that consists of a fixed set of $f + 1$ main acceptors and $f$ auxiliary acceptors. During failure-free execution (the normal case), only the main acceptors are contacted. The auxiliary acceptors perform MultiPaxos' horizontal reconfiguration protocol to replace failed main acceptors. As with Fast Paxos, we can deploy Matchmaker MultiPaxos with only $f + 1$ acceptors, $f$ fewer than Cheap Paxos. Matchmaker Paxos does require $2f + 1$ matchmakers, but matchmakers do not act as acceptors and have to process only a single message (i.e. a MATCHA message) to perform a reconfiguration.

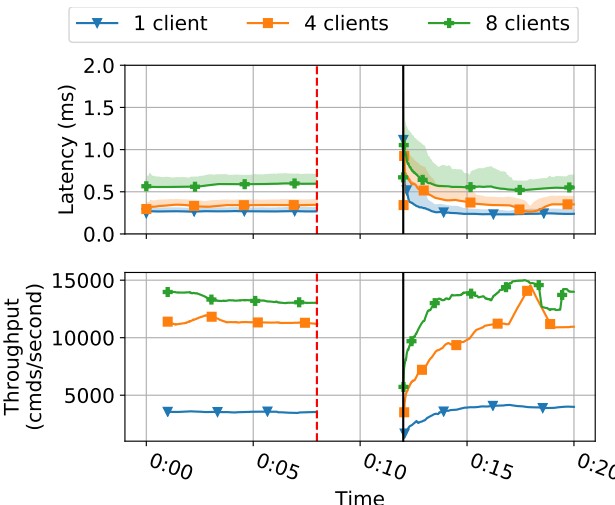

Figure 14: The latency and throughput of Horizontal Multi-Paxos with $f = 1$.

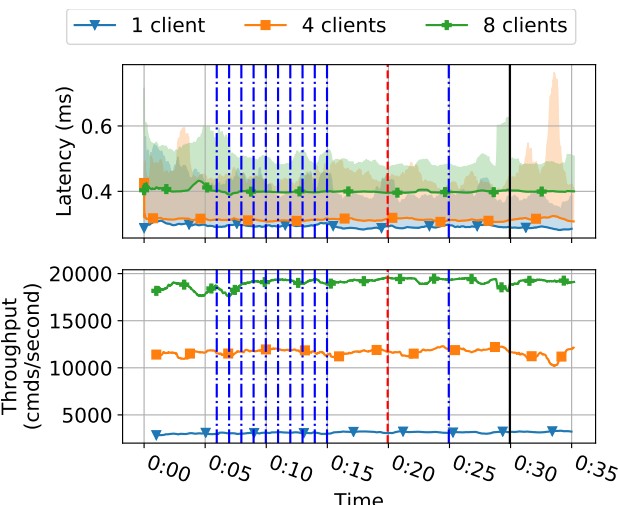

Figure 15: The latency and throughput of Matchmaker MultiPaxos ($f = 1$). The dotted blue, dashed red, and vertical black lines show matchmaker reconfigurations, a matchmaker failure, and an acceptor reconfiguration respectively.

**Raft.** Raft [35] uses a reconfiguration protocol called joint consensus. Like MultiPaxos' horizontal reconfiguration, joint consensus is log-based and therefore incompatible with many existing replication protocols. A simpler reconfiguration protocol for Raft was proposed in [34] but requires more rounds of communication.

**Viewstamped Replication (VR).** VR [26] uses a stop-the-world approach to reconfiguration. During a reconfiguration, the entire protocol stops processing commands. Thus, while the reconfiguration is quite simple, it is inefficient. Stoppable Paxos [21] is similar to MultiPaxos' horizontal reconfiguration, but also uses a stop-the-world approach. VR's stop-the-world approach to reconfiguration is also adopted by databases built on VR, including TAPIR [41] and Meerkat [39]. We use a similar approach to reconfigure matchmakers, but because matchmakers are off the critical path, the performance overheads are invisible.

**Fast Paxos Coordinated Recovery.** Fast Paxos has an optimization called coordinated recovery that is similar to Phase 1 Bypassing. The main difference is that in coordinated recovery, a leader uses Phase 2 information in round $i$ to skip Phase 1 in round $i+1$, whereas with Phase 1 Bypassing, the leader instead uses Phase 1 information. Note that coordinated recovery is not useful for Matchmaker MultiPaxos. It is subsumed by Phase 1 Bypassing. Coordinated recovery is only needed for Fast Paxos because the leader may not know which values were proposed in a round it owns. Phase 1 Bypassing cannot be applied to Fast Paxos for pretty much the same reason.

**DynaStore.** Vertical Paxos assumes its external master is implemented using state machine replication. MultiPaxos' horizontal reconfiguration also depends on consensus. Matchmaker Paxos does not require consensus to implement match-

makers, but we are not the first to notice this. DynaStore [1] showed that reconfiguring atomic storage does not require consensus.

**ZooKeeper.** ZooKeeper, a distributed coordinated service, which uses ZooKeeper Atomic Broadcast [14] is a protocol similar to MultiPaxos that can also reconfigure quickly after leader failures.

# 9 Conclusion

We presented Matchmaker Paxos and Matchmaker Multi-Paxos to address the lack of research on the increasingly important topic of reconfiguration. Our protocols achieve a number of desirable properties, both theoretical and practical: they can reconfigure without performance degradation, they provide insights into existing protocols, and they generalize better than existing techniques. Our implementations of Matchmaker Paxos and Matchmaker MultiPaxos can be found online at https://github.com/mwhittaker/frankenpaxos.

# Acknowledgement

This research is supported in part by DHS Award HSHQDC-16-3-00083, NSF CISE Expeditions Award CCF-1139158, NSF Award CNS-1815212, and gifts from Alibaba, Amazon Web Services, Ant Financial, CapitalOne, Ericsson, GE, Google, Huawei, Intel, IBM, Microsoft, Scotiabank, Splunk and VMware.

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

## A   Garbage Collection Safety

To prove that the three scenarios from Section 3.5 are safe, we repeat the safety proof from Section 3. The new bits are shown in blue.

*Proof.* We prove, for every round $i$, the statement $P(i)$: "if a proposer proposes a value $v$ in round $i$ (i.e. sends a PHASE2A message for value $v$ in round $i$), then no value other than $v$ has been or will be chosen in any round less than $i$." At most one value is ever proposed in a given round, so at most one value is ever chosen in a given round. Thus, $P(i)$ suffices to prove that Matchmaker Paxos is safe for the following reason. Assume for contradiction that Matchmaker Paxos chooses distinct values $x$ and $y$ in rounds $j$ and $i$ with $j < i$. Some proposer must have proposed $y$ in round $i$, so $P(i)$ ensures us that no value other than $y$ could have been chosen in round $j$. But, $x$ was chosen, a contradiction.

We prove $P(i)$ by strong induction on $i$. $P(0)$ is vacuous because there are no rounds less than 0. For the general case $P(i)$, we assume $P(0), \ldots, P(i-1)$. We perform a case analysis on the proposer's pseudocode (Algorithm 3). Either $k$ is $-1$ or it is not (line 11). First, assume it is not. In this

case, the proposer proposes $x$, the value proposed in round $k$ (line 12). We perform a case analysis on round $j$ to show that no value other than $x$ has been or will be chosen in any round $j < i$.

**Case 1:** $j > k$**.** We show that no value has been or will be chosen in round $j$. Recall that at the end of the Matchmaking phase, the proposer computed the set $H_i$ of prior configurations using responses from a set $M_i$ of $f + 1$ matchmakers. Either $H_i$ contains a configuration $C_j$ in round $j$ or it doesn't.

First, suppose it does. Then, the proposer sent PHASE1A$\langle i \rangle$ messages to all of the acceptors in $C_j$. A Phase 1 quorum of these acceptors, say $Q$, all received PHASE1A$\langle i \rangle$ messages and replied with PHASE1B messages. Thus, every acceptor in $Q$ set its round $r$ to $i$, and in doing so, promised to never vote in any round less than $i$. Moreover, none of the acceptors in $Q$ had voted in any round greater than $k$. So, every acceptor in $Q$ has not voted and never will vote in round $j$. For a value $v'$ to be chosen in round $j$, it must receive votes from some Phase 2 quorum $Q'$ of round $j$ acceptors. But, $Q$ and $Q'$ necessarily intersect, so this is impossible. Thus, no value has been or will be chosen in round $j$.

Now suppose that $H_i$ does *not* contain a configuration for round $j$. Either a configuration $C_j$ was garbage collected from $H_i$ or it wasn't. First, assume it wasn't. Then, $H_i$ is the union of $f + 1$ MATCHB messages from the $f + 1$ matchmakers in $M_i$. Thus, if $H_i$ does not contain a configuration for round $j$, then none of the MATCHB messages did either. This means that for every matchmaker $m \in M_i$, when $m$ received MATCHA$\langle i, C_i \rangle$, it did not contain a configuration for round $j$ in its log and never did. Moreover, by processing the MATCHA$\langle i, C_i \rangle$ request and inserting $C_i$ in log entry $i$, the matchmaker is guaranteed to never process a MATCHA$\langle j, C_j \rangle$ request in the future. Thus, every matchmaker in $M_i$ has not processed a MATCHA request in round $j$ and never will. For a value to be chosen in round $j$, the proposer executing round $j$ must first receive replies from $f + 1$ matchmakers, say $M_j$, in round $j$. But, $M_i$ and $M_j$ necessarily intersect, so this is impossible. Thus, no value has been or will be chosen in round $j$.

Otherwise, a configuration $C_j$ was garbage collected from $H_i$. Note that none of the matchmakers in $M_i$ had received a GARBAGEA$\langle i' \rangle$ command for a round $i' > i$ when they responded with their MATCHB messages. If they had, they would have ignored our MATCHA$\langle i, C_i \rangle$ message. Let $i'$ be the largest round $j < i' < i$ such that a matchmaker in $M_i$ had received a GARBAGEA$\langle i' \rangle$ message before responding to our MATCHA$\langle i, C_i \rangle$ message.

If $i'$ was garbage collected because of Scenario 1, then $k$ would be at least as large as $i'$ since we would have intersected the Phase 2 quorum of $C_{i'}$ used in round $i'$ to get a value chosen. But $k < j < i'$, a contradiction. If $i'$ was garbage collected because of Scenario 2, then we know no value has been or will be chosen in round $j$. If $i'$ was garbage collected because of Scenario 3, then we would have intersected the

Phase 2 quorum of $C_{i'}$ that knows a value was already chosen, and we would have not proposed a value in the first place. But, we proposed $x$, a contradiction.

**Case 2:** $j = k$**.** In a given round, at most one value is proposed, let alone chosen. $x$ is *the* value proposed in round $k$, so no other value could be chosen in round $k$.

**Case 3:** $j < k$**.** By induction, $P(k)$ states that no value other than $x$ has been or will be chosen in any round less than $k$. This includes round $j$.

Finally, if $k$ is $-1$, then we are in the same situation as in Case 1. No value has or will be chosen in a round $j < i$.  $\square$

## B  Matchmaker Reconfiguration Safety

We repeat the safety proof from Section A. The new bits are shown in blue.

*Proof.* We prove, for every round $i$, the statement $P(i)$: "if a proposer proposes a value $v$ in round $i$ (i.e. sends a PHASE2A message for value $v$ in round $i$), then no value other than $v$ has been or will be chosen in any round less than $i$." At most one value is ever proposed in a given round, so at most one value is ever chosen in a given round. Thus, $P(i)$ suffices to prove that Matchmaker Paxos is safe for the following reason. Assume for contradiction that Matchmaker Paxos chooses distinct values $x$ and $y$ in rounds $j$ and $i$ with $j < i$. Some proposer must have proposed $y$ in round $i$, so $P(i)$ ensures us that no value other than $y$ could have been chosen in round $j$. But, $x$ was chosen, a contradiction.

We prove $P(i)$ by strong induction on $i$. $P(0)$ is vacuous because there are no rounds less than 0. For the general case $P(i)$, we assume $P(0), \ldots, P(i-1)$. We perform a case analysis on the proposer's pseudocode (Algorithm 3). Either $k$ is $-1$ or it is not (line 11). First, assume it is not. In this case, the proposer proposes $x$, the value proposed in round $k$ (line 12). We perform a case analysis on round $j$ to show that no value other than $x$ has been or will be chosen in any round $j < i$.

**Case 1:** $j > k$**.** We show that no value has been or will be chosen in round $j$. Recall that at the end of the Matchmaking phase, the proposer computed the set $H_i$ of prior configurations using responses from a set $M_i$ of $f + 1$ matchmakers. Either $H_i$ contains a configuration $C_j$ in round $j$ or it doesn't.

First, suppose it does. Then, the proposer sent PHASE1A$\langle i \rangle$ messages to all of the acceptors in $C_j$. A Phase 1 quorum of these acceptors, say $Q$, all received PHASE1A$\langle i \rangle$ messages and replied with PHASE1B messages. Thus, every acceptor in $Q$ set its round $r$ to $i$, and in doing so, promised to never vote in any round less than $i$. Moreover, none of the acceptors in $Q$ had voted in any round greater than $k$. So, every acceptor in $Q$ has not voted and never will vote in round $j$. For a value $v'$ to be chosen in round $j$, it must receive votes from some Phase 2 quorum $Q'$ of round $j$ acceptors. But, $Q$

and $Q'$ necessarily intersect, so this is impossible. Thus, no value has been or will be chosen in round $j$.

Now suppose that $H_i$ does *not* contain a configuration for round $j$. Either a configuration $C_j$ was garbage collected from $H_i$ or it wasn't. First, assume it wasn't. Then, $H_i$ is the union of $f+1$ MATCHB messages from the $f+1$ matchmakers in $M_i$. Thus, if $H_i$ does not contain a configuration for round $j$, then none of the MATCHB messages did either. This means that for every matchmaker $m \in M_i$, when $m$ received MATCHA$\langle i, C_i \rangle$, it did not contain a configuration for round $j$ in its log and never did. Moreover, no majority in any previous set of matchmakers contained a configuration in round $j$. If any majority did have a configuration in round $j$, then all subsequent matchmakers would as well since a set of matchmakers is initialized from a majority of the previous matchmakers. Moreover, by processing the MATCHA$\langle i, C_i \rangle$ request and inserting $C_i$ in log entry $i$, the matchmaker is guaranteed to never process a MATCHA$\langle j, C_j \rangle$ request in the future. Moreover, no future set of matchmakers will either. A majority of matchmakers have a configuration in entry $i$, so all subsequent configurations will as well. Therefore, they will all reject a configuration in round $j$. Thus, every matchmaker in $M_i$ has not processed a MATCHA request in round $j$ and never will. For a value to be chosen in round $j$, the proposer executing round $j$ must first receive replies from $f+1$ matchmakers, say $M_j$, in round $j$. But, $M_i$ and $M_j$ necessarily intersect, so this is impossible. This argument holds for every set of matchmakers. Thus, no value has been or will be chosen in round $j$.

Otherwise, a configuration $C_j$ was garbage collected from $H_i$. Note that none of the matchmakers in $M_i$ had received a GARBAGEA$\langle i' \rangle$ command for a round $i' > i$ when they responded with their MATCHB messages. If they had, they would have ignored our MATCHA$\langle i, C_i \rangle$ message. Similarly, none of the matchmakers in $M_i$ were initialized with a garbage collection watermark $w > i$. Let $i'$ be the largest round $j < i' < i$ that a matchmaker in $M_i$ garbage collected before responding to our MATCHA$\langle i, C_i \rangle$ message.

If $i'$ was garbage collected because of Scenario 1, then $k$ would be at least as large as $i'$ since we would have intersected the Phase 2 quorum of $C_{i'}$ used in round $i'$ to get a value chosen. But $k < j < i'$, a contradiction. If $i'$ was garbage collected because of Scenario 2, then we know no value has been or will be chosen in round $j$. If $i'$ was garbage collected because of Scenario 3, then we would have intersected the Phase 2 quorum of $C_{i'}$ that knows a value was already chosen, and we would have not proposed a value in the first place. But, we proposed $x$, a contradiction.

**Case 2:** $j = k$. In a given round, at most one value is proposed, let alone chosen. $x$ is *the* value proposed in round $k$, so no other value could be chosen in round $k$.

**Case 3:** $j < k$. We can apply the inductive hypothesis to get $P(k)$ which states that no value other than $x$ has been or will be chosen in any round less than $k$. This includes round

---

**Algorithm 5** Fast Paxos Proposer Pseudocode

**State:** a round $i$, initially $-1$
**State:** the configuration $C_i$ for round $i$, initially null
**State:** the prior configurations $H_i$ for round $i$, initially null
1:  $i \leftarrow$ next largest round owned by this proposer
2:  $C_i \leftarrow$ an arbitrary configuration
3:  send MATCHA$\langle i, C_i \rangle$ to all of the matchmakers
4:  **upon** receiving MATCHB$\langle i, H_i^1 \rangle, \ldots,$ MATCHB$\langle i, H_i^{f+1} \rangle$ from $f+1$ matchmakers **do**
5:      $H_i \leftarrow \bigcup_{j=1}^{f+1} H_i^j$
6:      send PHASE1A$\langle i \rangle$ to every acceptor in $H_i$
7:  **upon** receiving PHASE1B$\langle i, -, - \rangle$ from a Phase 1 quorum from every configuration in $H_i$ **do**
8:      $k \leftarrow$ the largest $vr$ in any PHASE1B$\langle i, vr, vv \rangle$
9:      $V \leftarrow$ the corresponding $vv$'s in round $k$
10:     **if** $k = -1$ **then**
11:         send PHASE2A$\langle i, \mathsf{any} \rangle$ to every acceptor in $C_i$
12:     **else if** $V = \{v\}$ **then**
13:         send PHASE2A$\langle i, v \rangle$ to every acceptor in $C_i$
14:     **else**
15:         send PHASE2A$\langle i, \mathsf{any} \rangle$ to every acceptor in $C_i$

---

$j$, which is exactly what we're trying to prove.

Finally, if $k$ is $-1$, then we are in the same situation as in Case 1.  □

## C  Fast Paxos

Fast Paxos proposer pseudocode is given in Algorithm 5. We do not modify the Fast Paxos acceptor or the matchmakers. For simplicity, we assume that we deploy Fast Paxos with $f+1$ acceptors, with a single unanimous Phase 2 quorum, and with singleton Phase 1 quorums. Generalizing to arbitrary configurations that satisfy Fast Paxos' quorum intersection requirements is straightforward. Note that Fast Paxos cannot leverage Phase 1 Bypassing. Also note while both MultiPaxos and our Fast Paxos variant both have quorums of size $f+1$, our Fast Paxos variant has a *fixed* set of $f+1$ acceptors, while MultiPaxos can choose any set of $f+1$ acceptors from all $2f+1$ acceptors. This has some disadvantages in terms of tail latency and fault tolerance.

We now prove that our modifications to Fast Paxos are safe. For simplicity, we ignore garbage collection and matchmaker reconfiguration. Introducing those two features and proving them correct is pretty much identical to what we did with Matchmaker Paxos.

*Proof.* We prove, for every round $i$, the statement $P(i)$ which states that if an acceptor votes for a value $v$ in round $i$ (i.e. sends a PHASE2B message for value $v$ in round $i$), then no value other than $v$ has been or will be chosen in any round less than $i$. $P(i)$ suffices to prove that Matchmaker Paxos is

safe. Why? Well, assume for contradiction that Matchmaker Paxos chooses distinct values $x$ and $y$ in rounds $i$ and $j$ with $i < j$. Some acceptor must have voted for $y$ in round $j$, so $P(j)$ ensures us that no value other than $y$ could have been chosen in round $i$. But, $x$ was chosen, a contradiction.

We prove $P(i)$ by strong induction on $i$. $P(0)$ is vacuous because there are no rounds less than 0. For the general case $P(i)$, we assume $P(0), \ldots, P(i-1)$. We perform a case analysis on the proposer's pseudocode. Either $k$ is $-1$ or it is not (line 8). First, assume it is not. We perform a case analysis on rounds $j < i$.

**Case 1:** $j > k$. Recall that at the end of the Matchmaking phase, the proposer computed the set $H_i$ of prior configurations using responses from a set $M$ of $f + 1$ matchmakers. Either $H_i$ contains a configuration $C_j$ in round $j$ or it doesn't.

First, suppose it does. Then, the proposer sent PHASE1A$\langle i \rangle$ messages to all of the acceptors in $C_j$. A Phase 1 quorum of these acceptors, say $Q$, all received PHASE1A$\langle i \rangle$ messages and replied with PHASE1B messages. Thus, every acceptor in $Q$ set its round $r$ to $i$, and in doing so, promised to never vote in any round less than $i$. Moreover, none of the acceptors in $Q$ had voted in any round greater than $k$. So, every acceptor in $Q$ has not voted and never will vote in round $j$. For a value $v'$ to be chosen in round $j$, it must receive votes from some Phase 2 quorum $Q'$ of round $j$ acceptors. But, $Q$ and $Q'$ necessarily intersect, so this is impossible. Thus, no value has been or will be chosen in round $j$.

Now suppose that $H_i$ does *not* contain a configuration for round $j$. $H_i$ is the union of $f + 1$ MATCHB messages from the $f + 1$ matchmakers in $M$. Thus, if $H_i$ does not contain a configuration for round $j$, then none of the MATCHB messages did either. This means that for every matchmaker $m \in M$, when $m$ received MATCHA$\langle i, C_i \rangle$, it did not contain a configuration for round $j$ in its log. Moreover, by processing the MATCHA$\langle i, C_i \rangle$ request and inserting $C_i$ in log entry $i$, the matchmaker is guaranteed to never process a MATCHA$\langle j, C_j \rangle$ request in the future. Thus, every matchmaker in $M$ has not processed a MATCHA request in round $j$ and never will. For a value to be chosen in round $j$, the proposer executing round $j$ must first receive replies from $f + 1$ matchmakers, say $M'$, in round $j$. But, $M$ and $M'$ necessarily intersect, so this is impossible. Thus, no value has been or will be chosen in round $j$.

**Case 2:** $j = k$. If $V = \{v\}$, then the proposer proposes $v$. We must prove that no value other than $v$ has been or will be chosen in round $k$. For a value to be chosen in round $k$, every acceptor must vote for it in round $k$. Some acceptor voted for $v$ in round $k$, so it is the only value with the possibility of receiving a unanimous vote.

Otherwise $V$ contains multiple distinct elements, and the proposer proposes any. We must prove that no value has been or will be chosen in round $k$. This is immediate since no value can receive a unanimous vote in round $k$, if two different values have received votes in round $k$.

**Case 3:** $j < k$. If $V = \{v\}$, then the proposer proposes $v$, and we must prove that no value other than $v$ has been or will be chosen in any round less than $k$. This is immediate from $P(k)$. Otherwise, $V = \{v_1, v_2, \ldots\}$, and the proposer proposes any. We must prove that no value has been or will be chosen in any round less than $k$. $P(k)$ tells us that no value other than $v_1$ has been or will be chosen in any round less than $k$. $P(k)$ also tells us that no value other than $v_2$ has been or will be chosen in any round less than $k$. Thus, no value has been or will be chosen in any round less than $k$.

Finally, if $k$ is $-1$, then we are in the same situation as in Case 1. No value has been or will be chosen in any round less than $i$. $\square$

## D  DPaxos Bug

Consider a DPaxos deployment with $f_d = 1$, $f_z = 0$, three zones, three nodes per zone, and delegate quorums. Thus, a replication quorum consists of two nodes in one zone, and a leader election quorum consists of two nodes in two zones. We name the nodes $A$ through $I$. Beside each node, we display its ballot, vote ballot, vote value, and intents [20].

| Zone 1 | Zone 2 | Zone 3 |
|---|---|---|
| Ⓐ $-1, -1, \bot, \emptyset$ | | Ⓖ $-1, -1, \bot, \emptyset$ |
| | Ⓓ $-1, -1, \bot, \emptyset$ | |
| Ⓑ $-1, -1, \bot, \emptyset$ | | Ⓗ $-1, -1, \bot, \emptyset$ |
| | Ⓔ $-1, -1, \bot, \emptyset$ | |
| Ⓒ $-1, -1, \bot, \emptyset$ | | Ⓘ $-1, -1, \bot, \emptyset$ |
| | Ⓕ $-1, -1, \bot, \emptyset$ | |

Proposer 1 initiates the leader election phase in ballot 0 for value $x$. It selects $\{A, B, D, E\}$ as its leader election quorum and $\{B, C\}$ as its intent. It sends prepare messages to the leader election quorum, and the leader election quorum replies. Proposer 1 doesn't receive any intents, so it does not expand its leader election quorum. It also learns that no value has been chosen yet, so it proposes value $x$ to $B$ and $C$. Both accept the value.

| Zone 1 | Zone 2 | Zone 3 |
|---|---|---|
| Ⓐ $0, -1, \bot, \{0 : \{B, C\}\}$ | | Ⓖ $-1, -1, \bot, \emptyset$ |
| | Ⓓ $0, -1, \bot, \{0 : \{B, C\}\}$ | |
| Ⓑ $0, 0, x, \{0 : \{B, C\}\}$ | | Ⓗ $-1, -1, \bot, \emptyset$ |
| | Ⓔ $0, -1, \bot, \{0 : \{B, C\}\}$ | |
| Ⓒ $0, 0, x, \emptyset$ | | Ⓘ $-1, -1, \bot, \emptyset$ |
| | Ⓕ $-1, -1, \bot, \emptyset$ | |

Next, proposer 2 initiates the leader election phase in ballot 1 for value $y$. It selects $\{E, F, H, I\}$ as its leader election

quorum and $\{G,H\}$ as its intent. It sends prepare messages to the leader election quorum, and the leader election quorum replies. Proposer 2 receives the intent $\{B,C\}$ in ballot 0 from $E$, so it expands its leader election quorum and sends a prepare message to $C$. Proposer 2 learns that value $x$ was chosen in ballot 0, so it ditches $y$ and proposes $x$ to $G$ and $H$. $G$ accepts, but the propose message to $H$ is dropped.

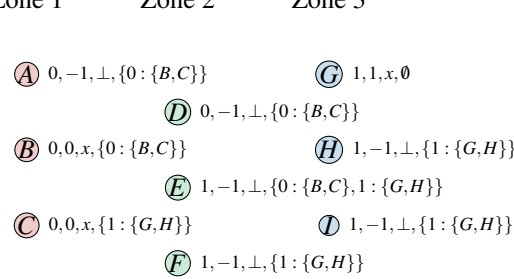

Next, garbage collection is run. The garbage collector contacts $G$ and sees that it has accepted a value in ballot 1. It informs all the nodes to discard intents in ballots less than 1.

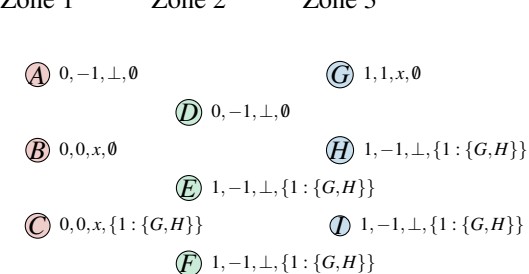

Next, proposer 3 initiates the leader election phase in ballot 2 for value $z$ It selects $\{D,E,H,I\}$ as its leader election quorum and $\{E,F\}$ as its intent. It sends prepare messages to the leader election quorum, and the leader election quorum replies. Proposer 3 receives intent $\{G,H\}$ in ballot 1, but has already included $H$ in its leader election quorum, so it does not send any additional prepares. It learns that no value has been chosen (this is a bug, $x$ was chosen), so it proposes value $z$ to $E$ and $G$. Both accept the value, and $z$ is chosen. This is a bug since $x$ was already chosen.

Zone 1    Zone 2    Zone 3

$A$ $0,-1,\perp,\emptyset$    $G$ $1,1,x,\emptyset$

$D$ $2,-1,\perp,\emptyset 2:\{E,F\}$

$B$ $0,0,x,\emptyset$    $H$ $2,-1,\perp,\{1:\{G,H\},2:\{E,F\}\}$

$E$ $2,2,z,\{1:\{G,H\},2:\{E,F\}\}$

$C$ $0,0,x,\{1:\{G,H\}\}$    $I$ $2,-1,\perp,\{1:\{G,H\},2:\{E,F\}\}$

$F$ $2,2,z,\{1:\{G,H\}\}$

