# OpenReview forum: "Solution: Matchmaker Paxos: A Reconfigurable Consensus Protocol"
_JSYS/2021/Mar_Papers — JSYS Mar 21_

### Official Review · AnonReviewer1 · 2021-03-31
**Informative paper on state machine replication with reconfiguration**

**Decision:**

Weak accept: good paper with flaws that can be fixed in three months

**Review:**

This paper presents Donut Paxos, a crash fault tolerant state machine replication system with vertical reconfiguration. The paper motivates vertical reconfiguration convincingly, provides useful details in the design of the systems, and evaluates the system with a prototype implementation. The paper is well written for the most part. I have a few suggestions for further improving the paper and clarifying certain issues.

I think the paper can help the readers by motivating vertical reconfiguration early on. While the introduction mentions the lack of logs in certain systems, it is too terse and easy to miss. This leaves me wondering about the motivation for vertical reconfiguration while reading the paper. I suggest the authors moving some of nice arguments from Section 6 to the introduction. In addition, it would be great if the authors can also comment on whether there are downsides with vertical reconfiguration and external matchmakers. I am also wondering if vertical reconfiguration is fundamentally tied to external masters, or is it just the only way we know. It would be helpful to see some discussion on this early on.

The paper contains fairly complete details for the most part. The only part I find a little lacking is matchmaker reconfiguration. Though the proof seems detailed, I need to see more details for the protocol to evaluate the proof. Running a Paxos instance makes sense intuitively, but how do proposers figure out current situation? For example, if one proposer shuts down the old matchmakers and immediately crashes, what do we do now? Another proposer may contact the old matchmaker and it will not hear back from them. How does it know if the old matchmakers have been shut down or if the messages are simply delayed? What's the criteria for this new proposer to perform a matchmaker reconfiguration? Does it directly install a new matchmaker configuration, or does it wait to hear back from the old matchmakers (which not never come because the old matchmakers have been shut down already)?

The experiments seem fair and thorough. My main question here is why more clients improve throughput. Is it because you assume each client generates commands at some rate and you need a certain number of them to saturate the system?  If that is the case, it is better to directly use the fundamental parameter of input command rate in the figure and I see no reason to use the number of clients as an arbitrary proxy. If it is for some other reason, it should be explained.  Please also explain the meaning of violin plots. I, for one, do not know what they are.


Below are some secondary comments on readability, clarity, and consistency.

Paxos works in a partially synchronous network, not an asynchronous one. As a deterministic protocol, it is subject to the FLP impossibility.

Section 3.4 mentions that premature garbage collection can make a proposer stuck; Section 3.5 instead says premature garbage collection can be unsafe. I think these are two sides of the same coin. It is better to be consistent about its effect. On a related note, I find it more natural to talk about when to garbage collect before how to garbage collect.

The "Paxos Made Simple" paper introduced three logical roles in Paxos: proposers, acceptors, and learners. The paper adopts the first two but not the third one. In fact, the paper seems to use "replicas" to refer to learners. I do not recommend this convention. To me, replicas refer to physical servers that may play multiple logical roles. To this end, I associate Paxos with 2f+1 replicas. Section 4.1 says there are f+1 replicas (learners in my interpretation) is a little confusing. Later the evaluation section says 2f+1 replicas, adding to the confusion.

While I agree that configuration selection is an orthogonal issue, I think it is helpful to fix a simple configuration selection mechanism and stick to it throughout the paper. Currently, Section 4.2 states that leader i selects its own configuration for round i but also mentions other alternatives like an external source; the first paragraph of Section 4.3 says that leader i selects the configuration for round i+1; the last paragraph of Section 4.3 reverts back to leader i selecting the configuration for round i.

Related to the above point, it is also helpful to fix a leader/proposer schedule. Section 4.2 suggests each round has a (possibly different) leader. But Section 4.4 seems to imply that the same leader is in charge of both round i and round i+1. The Phase 1 bypassing optimization seems to apply only when a single leader is in charge of both rounds. This makes it more important to discuss the leader schedule. If the same leader stays across rounds, then when do you change leaders? If you routinely change leaders, then you may need to discuss when Phase 1 bypassing is applicable, and revisit the claim that "no commands are delayed".

The title of section 6 is not the most suitable one. The section made several good insights, and generality is just one of them. Some of the points in the latter part of Section 6 may fit better in the related work section.






**Expertise:**

Actively publishing in this area

**Useful:**

yes

---

### Official Review · AnonReviewer2 · 2021-04-09
**Excellent idea, crisp definitions of the optimizations, some parts looks misleading**

**Decision:**

Weak accept: good paper with flaws that can be fixed in three months

**Review:**

Donut Paxos is a very good paper that proposes a solution to an actual problem (a reconfiguration of the consensus based systems) which covers both the most widespread log-based systems (multi-paxos, raft) and the promising alternative systems such as EPaxos.

It contains a lot of good practical optimizations and it will definitely benefit the community. However I recommend doing another iteration before accepting the paper because some of its parts look misleading or contradict each other.

The main idea, a reconfiguration protocol, is excellent. It takes from Vertical Paxos and then goes forward with replacing a master with a specialized linearizable replicated service which doesn't rely on consensus and has trivial reconfiguration protocol. Also Donut Paxos  improves the GC procedure. I find this approach much easier to understand and to reason about compared to horizontal variants such as joint consensus. One of the interesting implications of this approach is that it helps to get rid of the control messages which often looks like gaps to an application which uses consensus as a foundation for the  replicated log service.

Another great idea is the communication protocol between the proposers and the matchmaker services. It combines linearizable querying and updating into a single command, executes it with a single round trip and the protocol doesn't suffer from contention under concurrency (like dueling proposers do with Fast Paxos).

Last things I want to highlight is the formalization of the "phase 1 bypass" technique via the triple ballot numbers and the formalization of the pipelining via counting inflight requests (k). Those ideas (bypass, pipelining) have been around since "Paxos Made Simple" but often they're left underspecified and finding info on how to do it becomes a rite of passage for an engineer implementing a consensus based system.

What I find concerning is that the authors claim that reconfiguration takes only one round of communication. While being technically correct it feels misleading. Engineers measure duration of the reconfiguration from the moment it's started to the moment it's safe to shut down a node. Since the usual deployment of the consensus based systems in the industry consist of 2f+1` nodes with colocated roles (Etcd, Zookeeper, CockroachDB, Redpanda etc.) having the replicas running on the same nodes as acceptors requires to copy its state before turning a node off (to satisfy GC scenario #3). From this position the following statements "reconfiguring to a new set of machines takes one round trip of communication" and "all of our results hold in a co-located deployment as well" contradict each other.

Another issue is the underspecified replica reconfiguration protocol and the unclear responsibility of the replica nodes. The sentence "replicas can also be safely added or removed at any time so long as we ensure that commands replicated on f+1 replicas remain replicated on f+1 replicas" isn't aligned with "state machine replicas execute the commands in log order". After reading them it's unclear whether replicas store commands after execution and how long they should do it.

Next issue is related to determinism of Paxos rounds: "every round is orchestrated by a single predetermined proposer". This sentence creates an impression that given a current round (a ballot number) it's possible to predict the next one but it isn't true for example after (4,"p1") there may be (4,"p2"), (5,"p1") or any other ballot number which is lexicographically greater than the current round.

A related thing is using `i+1` notation for the next round. The next round (ballot number) isn't defined until it happens so defining it as a function from the previous round isn't right. Also operation `+1` isn't defined on tuples ("let the set of rounds be the set of lexicographically ordered integer pairs (r,id) where r is an integer and id is a unique proposer id"). It might be better to use "next round" instead of `i+1` or to use subscripts `r_i` and `r_{i+1}` instead of `i` and `i+1`.

The last disturbing part is the evaluation section: the experiment is subject to the coordinated omission problem, it's unclear why the median and IQR metrics were chosen and the conclusion is self contradicting.

Clients wait to receive response before issuing the next command. This setup is subjected to the coordinated omission problem, a phenomenon when the measuring system coordinates with the measured system resulting in hiding the long periods of unavailability behind a single outlier which may be ignored even if we look at p99.

The evaluation part lacks a discussion why median and IQR are the important metrics. Even when the experiment is set up to avoid coordinated omission the reconfiguration still may significantly delay the requests without affecting the chosen metrics. For example when reconfiguration happens every second, lasts 250ms (a quarter of a second) and blocks the overlapping requests it doesn't affect IQR (a function from p75) but delays the requests by 850% (from 0.29ms to 250ms).

The following statements from the evaluation section contradict each other:

 - "we include the comparison to MultiPaxos for the sake of having some baseline against which we can compare Donut MultiPaxos, but the comparison is shallow"

- "Donut MultiPaxos does provide performance benefits over MultiPaxos’ and Raft’s reconfiguration protocols"

I recommend to redesign the experiment to avoid the coordinated omission problem by maintaining the constant rate of the requests, come up with meaningful metrics (e.g. to compare statistics like p99 and max latency of the requests overlapping with the reconfiguration and not overlapping) and back up performance benefits of Donut Paxos with numbers.

**Expertise:**

Follow the literature closely, last published 5+ years ago

**Useful:**

yes

---

### Official Review · AnonReviewer3 · 2021-04-10
**I think the paper is addressing and interesting problem. However, I have some concerns regarding the novelty of the work and the evaluation.**

**Decision:**

Weak reject: interesting papers with flaws, not sure if they can be fixed in three months

**Review:**

**Summary of the paper**

The paper presents Donut Paxos and Donut MultiPaxos which are a reconfigurable consensus protocol and reconfigurable replicated state machine, respectively. The primary value of the proposed protocols that they allow quick reconfiguration of the active set of acceptors while having no performance degradation. The proposed protocol performs reconfiguration off the critical path of standard command processing by employing a set of matchmaker processes. Matchmakers implement a log that stores the current and previous configurations (i.e., history of all configurations). Also, Donut Paxos/MultiPaxos uses vertical reconfiguration instead of the classical horizontal configuration that is used in Paxos. Vertical reconfiguration enables the system to perform reconfiguration quickly without the need to wait for the execution of some user commands. The evaluation results show that Donut Paxos/MultiPaxos performs reconfiguration quickly (orders of milliseconds) and does not have any overhead on the systems performance.


**Comments for authors**

The paper addresses an interesting problem that is usually poorly discussed in papers proposing consensus protocols. I like the way in which the solution was presented starting from building a reconfigurable consensus protocol (Donut Paxos) and then extending it to build a reconfigurable replicated state machine (Donut MultiPaxos). Also, I commend the authors for having a detailed proof of the safety of the protocol as well as for discussing how other consensus protocols can be extended to implement vertical reconfiguration.

However, I mark this paper as weak reject for several reasons. First, I have some concerns about the novelty of the solution; you can think of the matchmakers as a separate state machine that is used to manage the configurations. That is, rather than having a single state machines for both user commands and configurations, Donut MultiPaxos decouples them and has one for each. Moreover, there are 2f+1 matchmakers and a proposer waits for f+1 (typical majority) matchmaker responses to write current set of acceptors and get all previous configurations.

Second, this decoupling between configuration and command processing accelerates configuration changes. However, it increases the probability of unavailability as the availability of the commands’ state machine is tightly coupled to the availability of the matchmakers’ state machine. That is, if the matchmakers’ state machine is not available, a new leader/proposer cannot execute any command even if all acceptors are up and running. The paper does not discuss this issue or evaluate it.

Third, the paper addresses the reconfiguration of the acceptors only and do not discuss the reconfiguration of replicas. That is, the only place in which the paper discusses that is in the following excerpt:
“Replicas can also be safely added or removed at any time so long as we ensure that commands replicated on f +1 replicas remain replicated on f +1 replicas.”
However, I believe that the issue of making sure that commands are always replicated on majority of replicas is very tricky and this is what guarantees the safety of a replicated state machine.

Fourth, Donut Paxos was not rigorously evaluated. For instance only 8 clients were used which is very small number of clients and not enough to stress the system. Also, the experiments do not highlight any benefits for Donut MultiPaxos over MultiPaxos in terms of throughput and latency. For better evaluation and better presentation of results, I suggest having a throughput-latency graph that compares Donut MultiPaxos and MultiPaxos for different number of clients and different alpha values (i.e., concurrent commands). The following are other minor notes I have about the experiments:
- The results shown in Figure 9 and 10 are confusing; why Donut MultiPaxos has higher throughput that MultiPaxos in the first 10 seconds (no reconfigurations occur)?
- Figure 14 and its caption does match the text used to describe it in section 7.2.
- Having Donut MultiPaxos and MultiPaxos results in the same Figure will make it easier to read and compare results.

Finally, a minor point, I found that the text is not accurate is some places:
- “Many state machine replication protocol do not have logs and cannot perform horizontal reconfiguration [2, 8, 27, 30, 33]” (Introduction). The citations include EPaxos and Raft and both have logs.
- Sections 2.2 states that Paxos requires f+1 proposes, and I think it does not, and it works fine with any number of proposers.
- In Figure 4, should the number of replicas be 2f+1?
- “Raft cannot safely perform horizontal reconfigurations” (Section 6). I believe that Raft reconfiguration is based on the replicated log.
- “protocol. However, while none of the protocols have logs” (Section 6). This not accurate.




**Expertise:**

Published in this area in the last 5 years

**Useful:**

yes

---

### Official Review · AnonReviewer4 · 2021-04-11
**Donut Paxos: a Reconfigurable Consensus Protocol**

**Decision:**

Weak accept: good paper with flaws that can be fixed in three months

**Review:**

Summary

Donut Paxos is a variant of MultiPaxos that uses separate nodes for configuration management. It addresses the problem of missing reconfiguration protocols in academic consensus protocols. Performance is on par with horizontal MultiPaxos and Raft.

Strength
- This paper focuses on the reconfiguration problem. It may give more insights on reconfiguration to readers.

Weakness
- Weak motivation.
- The paper's solution is missing an important component.

I got many questions after reading through this paper. The primary concern is that this paper is missing a good motivation. I agree that reconfiguration is an important piece for a consensus protocol to be practical. But in the industry, consensus protocols are mostly used for configuration service, which manages replicas of primary backup replication. Even for the case when Paxos-like consensus protocol is used for main applications, replacing a failed server can be handled by the network layer (e.g., updating DNS to redirect requests to a new server). Or simple horizontal Paxos-like approach may work fine. To motivate Donut Paxos well, there should be some explanation on why it is important to provide reconfiguration service directly within consensus protocols.

Another problem is Donut Paxos doesn't provide a complete solution to the reconfiguration problem in state machine replication. Donut Paxos only covers how to safely reconfigure Paxos acceptors and ensure that servers can continue to reach an agreement. However, the difficulty of reconfiguration comes with recovering the state machine on a new replica. When Google first implemented MultiPaxos, they struggled a lot with the subtleties with recovering state machines. (Lamport's paper doesn't cover this topic.) On the other hand, Raft could be adopted widely in the industry since it fully specified how to handle the recovery of state machines, including recovery from a snapshot (they are discussed in Diego's PhD thesis). Without disclosing how to recover the state machine safely, Donut Paxos has the same problem as horizontal MultiPaxos, which will prevents the industry's adoption again.

Lastly, the current introduction implies that the exiting reconfiguration protocols cause performance degradation and that Donut Paxos has superior performance to them. However, in the evaluation section, Multi-Paxos's performance is the same as Donut Paxos's. It was a surprise when the evaluation section admits that the performance is not the contribution of Donut Paxos. I think this should be made clear earlier.

My suggestions are
- Clarify contribution: I think Donut Paxos can be valuable if it can be applied to any Paxos-based protocols. To show that, I think it should show another example, maybe applying Donut Paxos to EPaxos. It's a bit disappointing to see just the Multi-Paxos example as it already has horizontal reconfiguration.
- Extend protocol to the full solution. If you believe that missing reconfiguration was the reason why the industry didn't adopt improved Paxos variants, you should provide the full solution.
- I would conduct a survey on how industry people use consensus protocols and what problems they are suffering from. Then, Donut Paxos can be motivated more strongly.

Miscellaneous comments
- Page 1. "inefficient reconfiguration protocols [15, 22]" ==> can you evaluate their performance and show Donut Paxos's performance advantage?
- Page 1. "4% effect on the median of throughput and latency measurements" => what does it mean? Median of the whole experiment with a failure? If that's the case, medians cannot capture the reconfiguration behavior. If this is referring to the experiment in Section 7.1, the paper said reconfiguration completes within a millisecond, and the median was captured within one second window. I think that the "4%" number will change if a different time window was chosen.
- Page 3. "MATCHA" and "MATCHB" are a bit confusing.
- Page 4. "The proposer then ends the Matchmaking phase" ==> Is there a termination guarantee?
- Page 4. "to every acceptor in every configuration in H_i" ==> Isn't it just 1 per round?
- Page 4. In Algorithm 3, "an arbitrary configuration" ==> is it truly arbitrary?
- Page 4. In Algorithm 3, "Phase 1 quorum from every configuration in H_i" ==> what is the size of quorum?

**Expertise:**

Published in this area in the last 5 years

**Useful:**

yes

---

### Meta-Review · Area_Chair1 · 2021-04-15

**Recommendation:** Revise
**Confidence:** 5

**Metareview:**

Dear Authors, thank you for submitting to JSys. After reading the reviews, I recommend a REVISE decision. I believe the concerns of reviewers can be addressed in the next three months. After the submission of your revised version, reviewers will evaluate your modifications and a final decision will be made.

In general, most reviewers are mildly optimistic for a future acceptance of the paper. I believe all concerns can be addressed during the paper revision. Reviewers have included suggestions for improving the paper, please consider them.

Isolating some major concerns:
- Lack of practical motivation of why reconfiguration is critical to be integrated into the consensus protocol and not offloaded to an external service.
- Missing details about reconfiguration protocol.
- The performance evaluation should be improved. Multiple reviewers pointed out questionable decisions in terms of experimental settings and metrics.

We look forward to receiving your revised version.
Thank  you one more time for submitting,

Roberto Palmieri
Area co-chair of JSys

---

### Decision · Program_Chairs · 2021-04-15

**Decision:**

Accept

**Comment:**

The revision addressed most of the concerns of the reviewers. The remaining concern(s) can be fixed in shepherding.